# Scalable GANs with Transformers

**Sangeek Hyun** [1]  **Minkyu Lee** [1]  **Jae-Pil Heo** [1]

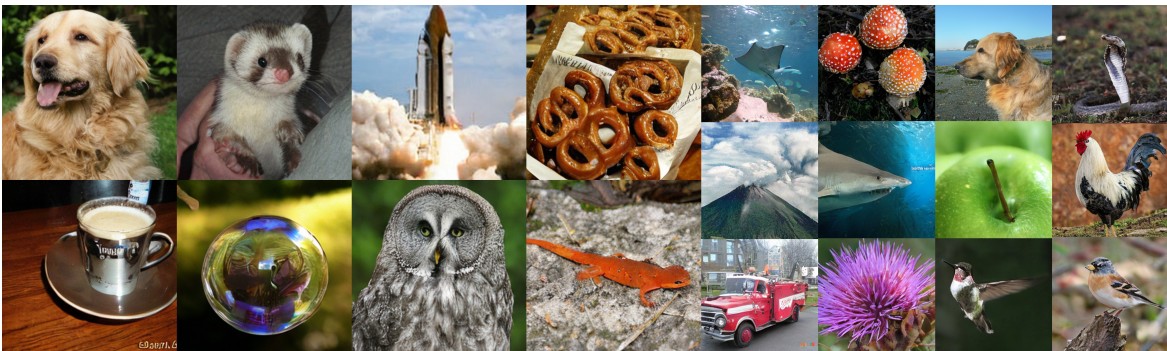

*Figure 1.* **Curated examples of GAT-XL/2 on ImageNet-256.** GAT-XL/2 exhibits strong generation capability (FID 2.18) within 60 epochs, 4× fewer than 1-NFE baselines (FID 3.43), while keeping the characteristics of GANs such as latent interpolation. More visualizations are available in Appendix.

## Abstract

Scalability has driven recent advances in generative modeling, yet it remains underexplored for adversarial learning. We study the scaling behavior of Generative Adversarial Networks through two design choices: training in a compact Variational Autoencoder latent space and using purely transformer-based generators and discriminators. While this setup is efficient and scales well with compute, naively scaling exposes failure modes; underutilization of early layers in the generator and increasing optimization instability. We address these issues with lightweight intermediate supervision and width-aware learning-rate adjustment. Our Generative Adversarial Transformers (GAT) train reliably from small (S) to extra-large (XL) model sizes, and GAT-XL model achieves state-of-the-art single-step class-conditional generation on ImageNet at 256×256 resolution (FID of 2.18) in 60 epochs, requiring 4× fewer epochs than strong baselines. Project page: https://hse1032.github.io/GAT.

## 1. Introduction

Recent breakthroughs in generative modeling have become a central driver of progress across core areas of computer vision. These developments have accelerated in recent years, enabling capabilities that were previously out of reach: state-of-the-art systems now support text-to-image (Rombach et al., 2022; Podell et al., 2024; Esser et al., 2024b; Han et al., 2025) and text-to-video synthesis (Yang et al., 2025; Chen et al., 2024; Bar-Tal et al., 2024), demonstrate practical applications (Brooks et al., 2024; Google DeepMind, 2025c;b), and further enable the creation of 3D content (Zhao et al., 2025) and large-scale world simulation models (Google DeepMind, 2025a).

At the core of this advance is scalability: enlarging model capacity and data coverage reliably improves performance, often near-monotonically. When pushed to sufficiently large regimes, these trends yield marked gains in fidelity, coverage, and controllability. Crucially, these benefits depend on scale-friendly choices, including architectures that maintain stable signal flow, training recipes that remain well-behaved as width, depth, and batch size grow, and computational efficiency. Such scaling behavior has already been demonstrated in certain types of generative models such as autoregressive and diffusion families (Tian et al., 2024; Peebles & Xie, 2023; Liang et al., 2026).

By contrast, the scalability of Generative Adversarial Networks (GANs) has not been discussed yet, despite its attractive single-step sampling efficiency and interesting property

[1]Sungkyunkwan University, Suwon, Republic of Korea. Correspondence to: Jae-Pil Heo <jaepilheo@skku.edu>.

*Proceedings of the 43rd International Conference on Machine Learning*, Seoul, South Korea. PMLR 306, 2026. Copyright 2026 by the author(s).

of semantic latent space. While there have been attempts to train GANs at large scale (Kang et al., 2023; Zhu et al., 2025; Sauer et al., 2023), these efforts typically focus on a single high-capacity model with extensive, task-specific tuning, and thus do not constitute evidence of genuine scalability.

In this work, we revisit GANs from a scalability perspective. We focus on two ingredients central to successful scalable generative models. First, these models are typically trained in a low-dimensional latent space; the spatial latent grid produced by a pretrained, frozen VAE (Rombach et al., 2022) as an image tokenizer/de-tokenizer, enabling a dramatic reduction of the computational burden of learning and inference while preserving high perceptual fidelity. Second, they employ transformer architectures, known for scalability against width, depth, data, and compute.

Inspired by these two crucial factors, we combine them to build a novel, scalable GAN framework: we construct a pure transformer-based GAN that operates in a compact latent space and study its behavior across substantial capacity ranges. We aim to assess the scalability of this design and to clarify the architectural and optimization choices. Accordingly, we pinpoint the hurdles that hinder adversarial training at scale. In detail, we identify two key problems: (1) the early layers of the generator become inactive, leading to marginal contribution in image synthesis and (2) naïvely increasing depth and width with identical configuration (e.g., same learning rate) leads to failures in convergence.

To address the first issue, we propose Multi-level Noise-perturbed image Guidance (MNG), which provides supervision at multiple intermediate layers of the generator. Specifically, we leverage a noise hierarchy: the synthesized images from earlier stages are trained to resemble the real data perturbed by a stronger image-level Gaussian, and the noise level monotonically decreases with depth. They serve as direct supervision for the generator's intermediate layers, restoring early-layer influence and improving layer-wise utilization throughout the network, and also encourage a coarse-to-fine refinement behavior across depth.

For the second issue, we note that as the model becomes deeper and wider, standard initialization and optimization tend to induce scale-dependent output updates. In particular, larger models often exhibit larger per-step output changes under the same learning rate, meaning that the generator and discriminator outputs can drift more rapidly at each optimization step. This effectively makes the training "speed" depend on model scale, which can destabilize GAN training dynamics already known for their sensitivity to hyperparameters (Lucic et al., 2018). To mitigate this, we introduce a simple scaling rule, especially for the learning rate, designed to keep the magnitude of per-step output changes approximately consistent across different model sizes.

We experimentally validate that our framework, Generative Adversarial Transformers (GAT), is successfully trained on various scales of model (GAT-S to GAT-XL) and achieves FID of 2.18, which is the state-of-the-art performance in a one-step generation task on the class-conditional generation in ImageNet-256 dataset only within 60 epochs of training, while keeping the advantages of GAN, such as a single inference step or latent space manipulation (Fig. 1, more examples are available in Appendix).

## 2. Proposed Method

### 2.1. Preliminaries

**Generative Adversarial Networks** Generative Adversarial Networks (GAN) (Goodfellow et al., 2014) is an adversarial learning framework between two networks, the generator $G(z, c)$ and discriminator $D(I, c)$. Specifically, for a given randomly sampled latent code $z \in \mathbb{R}^{d_z} \sim p_z$ and condition $c$, the generator $G(z, c)$ synthesizes a fake image $\hat{x} \in \mathbb{R}^{H \times W \times 3}$ and the discriminator learns to distinguish the real image $x \in \mathbb{R}^{H \times W \times 3}$ and the fake image $\hat{x}$, while the generator learns to deceive the discriminator.

GANs have several appealing properties compared to diffusion and AutoRegressive (AR) models: they use a very low-dimensional, semantically meaningful latent space (e.g., $d_z = 64$) that supports image manipulation, and they generate images in a single forward pass, enabling highly efficient inference. Despite these advantages, GAN scalability remains underexplored—unlike other modern generative models. In this work, we investigate how to scale GANs using a Transformer-based architecture with proven scalability across tasks.

### 2.2. Generative Adversarial Transformers

We introduce Generative Adversarial Transformers (GAT), a transformer-based GAN framework trained in the VAE latent space. While Transformers are widely regarded as highly scalable backbones, their use in GANs has been explored only to a limited extent, especially in the aspect of scalability. As a first step toward a scalable GAN with minimal architectural deviation, we build GAT as a *pure Transformer* design and demonstrate its feasibility at scale.

Specifically, we operate in latent space of a pre-trained VAE (Rombach et al., 2022), following recent advances in generative modeling (Rombach et al., 2022; Peebles & Xie, 2023; Tian et al., 2024). Training in the VAE latent space substantially reduces computational burden, enabling efficient scaling of both model size and training. For simplicity, we use terms "VAE latent" and "image" interchangeably.

**Generator architecture** Our generator adopts a standard Vision Transformer (ViT) architecture, consisting primarily

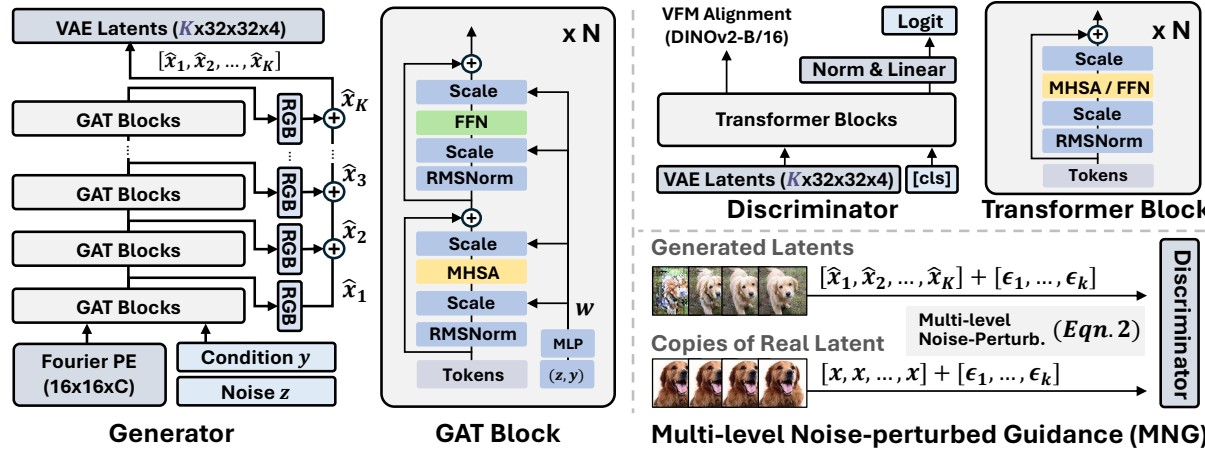

*Figure 2.* Generative Adversarial Transformers (GAT) architecture. Both generator and discriminator are built from transformer blocks, augmented with modulation in $G$ and Layerscale in $D$. Our generator synthesizes auxiliary outputs from intermediate layers, which are paired with multiple noise levels and forwarded into the discriminator. Through supervision on intermediate outputs, Multi-level Noise-perturbed Guidance (MNG) encourages all layers to contribute to images and leverages the model capacity more effectively.

of a stack of transformer blocks. Since the generator does not take input images, we remove the patchify layer and instead introduce an unpatchify layer (i.e., the RGB layer in Fig. 2) to synthesize images. Specifically, the unpatchify layer acts as a linear decoder, comprising normalization, linear projection, and reshaping operations. The output dimension of this linear decoder scales with the patch size $p$, increasing proportionally to $p^2$.

The transformer block (GAT block) follows a standard ViT-style design, augmented with conditioning on the latent code $z$ and class label $c$. We use a lightweight mapping MLP to produce a style vector $w$ from $(z, c)$, and generate per-channel scaling parameters $(\gamma, \alpha)$ from $w$ to modulate the block via adaptive normalization and LayerScale (Touvron et al., 2021). As RMSNorm is adopted as normalization, we omit the shift term. For stability, $\gamma$ and $\alpha$ are initialized to near-zero values and learned during training.

Note that, we call this generator architecture a *pure Transformer*: the backbone strictly follows ViT without convolution, and the additional component (e.g., adaptive normalization) is limited to lightweight feature modulation.

**Discriminator architecture** The discriminator also adopts a Vision Transformer (ViT) backbone, with Layerscale applied to each transformer block output. As in the generator, Layerscale parameters are initialized to small values to ensure stability during early training. For real/fake classification, a dedicated [cls] token is appended to the sequence of visual tokens before the first transformer block. This [cls] token is processed jointly with the other tokens and then passed through a linear projection head to produce the discriminator logit.

### 2.3. Activating Early Generator Layers via Multi-Level Noise-Perturbed Image Guidance

One key prerequisite for scalability is efficient model utilization: if a large portion of the network remains effectively inactive, scaling up yields limited benefits. In our experiments, we observe that Transformer-based generators, especially early blocks, are under-utilized. Specifically, during image synthesis, we observe several consistent pieces of evidence that collectively indicate under-utilization of the early generator blocks: (i) early-layer feature visualizations exhibit only marginal structural variation (Fig. 4a); (ii) ablating early blocks results in only minor changes in the generated images (Fig. 4b); and (iii) early representations appear largely insensitive to the latent code and are nearly unchanged across individual block transformations (Fig. A.7).

To ensure the effective scaling of GANs, we propose the Multi-level Noise-perturbed image Guidance (MNG) strategy for training GANs. Firstly, we divide the generator into multiple $K$ stages and enforce auxiliary outputs at each stage. Each intermediate output is connected to the final synthesis path through residual connections, ensuring that information from early blocks is not discarded but accumulated across depth. Throughout this process, for the intermediate output $\hat{x}_k$ at $k^{\text{th}}$ stage, the output of the generator is defined as follows:

$$G(z, c) = [\hat{x}_1, \hat{x}_2, ..., \hat{x}_k].\qquad(1)$$

Then, we perturb each intermediate output $x_k$ by a Gaussian noise with a predefined noise strength. In detail, the predefined strengths build a hierarchy by assigning stronger noise perturbation to earlier stages and weaker corruption

to later ones. After perturbation, all perturbed images are forwarded to the discriminator, guiding each generator stage to learn only the level of coarse structure that survives under its pre-defined noise. This process is defined as follows:

$$\mathcal{E}(\hat{x}_k) = \alpha_k \hat{x}_k + \sqrt{1 - \alpha_k^2}\, \epsilon, \qquad \epsilon \sim \mathcal{N}(0, I), \quad (2)$$

$$\alpha_1 < \alpha_2 < \cdots < \alpha_K, \qquad \alpha_K = 1, \quad (3)$$

$$\ell = D(\mathcal{E}([\hat{x}_1, ..., \hat{x}_k]), c) = D([\mathcal{E}(\hat{x}_1), ..., \mathcal{E}(\hat{x}_k)], c) \,(4)$$

where $\ell$ is logit and $\hat{x}_k$ is the noised-perturbed counterpart of $x_k$ and $\alpha_k$ controls the degree of perturbation for noise-level $k$, increasing exponentially with depth. For simplicity, we omit the noise strength $a_k$ for noise perturbing operator $\mathcal{E}$. Thus, earlier layers are supervised to match heavily noised images ($x_1$), while later layers are aligned with clean targets ($x_K$), forming a coarse-to-fine trajectory. For real data $x$, we use identical images for every level $k$.

This strategy encourages the early layers to capture global structure under strong noise corruption, while later layers progressively refine fine-grained details as the noise diminishes. Thus, it ensures that all layers contribute actively to the synthesis process, mitigating the problem of inactive early layers, which potentially leads to limited performance at scale. This is conceptually related to MSG-GAN (Karnewar & Wang, 2020), but it does not require an explicit image hierarchy; instead, we associate a single image with multiple noised counterparts via noising, which we expect to reduce shortcuts of discriminating real and fake by cross-scale consistency. Importantly, this mechanism incurs only negligible computational overhead while improving network utilization, especially in early layers.

### 2.4. Scaling rule for stabilizing the training of GAN

Recent diffusion models such as DiT (Peebles & Xie, 2023) demonstrate scalability while adopting identical hyperparameters regardless of model size. In contrast, we find that simply increasing the model size under an identical configuration often leads to training divergence in GANs. This is problematic as the manual tuning of hyperparameters for every scale would severely undermine scalability. To address this, we propose a simple and principled scaling rule.

The key idea of the guiding principle is to maintain a consistent update magnitude across different model widths. In practice, when each layer input is normalized to unit variance (as ensured by normalization layers), the expected squared norm of the input grows linearly with the number of channels. Consequently, the update rate of the model becomes proportional to both the learning rate and the channel dimension. Since GAN training is known to be highly unstable and particularly sensitive to the choice of learning rate (Lucic et al., 2018), preserving a constant update magnitude is crucial for preventing divergence and ensuring stable adversarial training dynamics. Therefore, when scaling up

the model size, the learning rate should decrease inversely with the number of channels so that the overall update scale remains stable.

Formally, let $\eta_{base}$ denote the learning rate for the *base* model with channel size $C_{base}$, where the *base* model is the model that we tune the hyperparameters. For a model with channel size $C_{\mathrm{model}}$, we define the learning rate adapted for this model $\eta_{\mathrm{adapt}}$ as follows:

$$\eta_{\mathrm{adapt}} = \eta_{base} \cdot \frac{C_{base}}{C_{\mathrm{model}}}. \quad (5)$$

We also provide mathematical justification of this claim in Appendix A.10.

### 2.5. Training objectives

For adversarial learning, we deploy relativistic pairing loss (Jolicoeur-Martineau, 2019) with two-sided gradient penalty following R3GAN (Huang et al., 2024), but an approximated version of it (Lin et al., 2025). Specifically, this objective is denoted as follows:

$$\mathcal{L}_G^{\mathrm{adv}} = f(D(\mathcal{E}(G(z, c)), c) - D(\mathcal{E}(x), c)), \quad (6)$$

$$\mathcal{L}_D^{\mathrm{adv}} = f(D(\mathcal{E}(x), c) - D(\mathcal{E}(G(z, c)), c)), \quad (7)$$

$$\mathcal{L}_{\mathrm{aR1}} = \tfrac{1}{\sigma^2} \left\| D(\mathcal{E}(x), c) - D(\mathcal{E}(x + \epsilon'), c) \right\|^2, \quad (8)$$

$$\mathcal{L}_{\mathrm{aR2}} = \tfrac{1}{\sigma^2} \left\| D(\mathcal{E}(G(z, c)), c) - D(\mathcal{E}(G(z, c) + \epsilon'), c) \right\|^2, \quad (9)$$

where $f(\cdot)$ is a softplus function and $\epsilon' \sim \mathcal{N}(0, \sigma I)$ is a gaussian noise with a std $\sigma$. Note that, $x$ and $G(z, c)$ are multi-level noise-perturbed images, where we omit them here for simplicity.

**Representation alignment on discriminator.** In addition, inspired by the rationale of feature-aided GANs (Sauer et al., 2021; Kumari et al., 2022) and recent diffusion work on representation alignment (Yu et al., 2025), we encourage the discriminator to learn semantically rich Vision Foundation Models (VFM) features. Different from prior work (Yu et al., 2025), we do not use the generator for alignment, as $G$ takes noise as input and it is difficult to obtain VFM features directly from the generated (fake) data. Let $\phi(\cdot)$ be a frozen vision foundation model (e.g., DINOv2 (Oquab et al., 2024)), and let $H_D(x) = \{h_{\mathrm{cls}}, h_1, \ldots, h_N\}$ denote the discriminator's [cls] token and $N$ patch tokens at the last layer. We obtain teacher tokens $\hat{H}_\phi(x) = \{\hat{h}_{\mathrm{cls}}, \hat{h}_1, \ldots, \hat{h}_N\}$ by forwarding the same image through $\phi$. Then, this alignment objective is defined as follows:

$$\mathcal{L}_{\mathrm{REPA}} = \frac{1}{N+1} \sum_{i \in \{\mathrm{cls}, 1:N\}} (\mathrm{sim}(P(h_i), \hat{h}_i)). \quad (10)$$

Note that, this alignment objective is only applied with a real data, and $P$ denotes a small learnable MLP to align

*Table 1.* **Class-conditional generation on ImageNet-256×256 (FID-50K)**. (Left) 1 or 2 Number of Function Evaluation (NFE) generative models. (Right) Other generative models including autoregressive models and multi-step diffusion/flow models. Diffusion/flow entries are reported under CFG, when applicable. Across both tables, '×2' denotes that CFG yields 2 NFEs for each sampling step. †: Leveraging ImageNet-pretrained discriminators, lowering FID more than the actual image quality (Kynkäänniemi et al., 2023a).

| Method | Params | NFE | Epoch | FID | | Method | Params | NFE | FID |
|---|---|---|---|---|---|---|---|---|---|
| ***2-NFE diffusion/flow from scratch*** | | | | | | ***autoregressive/masking*** | | | |
| iCT-XL/2 | 675M | 2 | - | 20.30 | | AR w/ VQGAN | 227M | 1024 | 26.52 |
| iMM-XL/2 | 675M | 1×2 | 3840 | 7.77 | | MaskGIT | 227M | 8 | 6.18 |
| MeanFlow-XL/2 | 676M | 2 | 240 | 2.93 | | STARFlow | 1.4B | 1024×2 | 2.40 |
| ***1-NFE diffusion/flow from scratch*** | | | | | | VAR-$d$30 | 2B | 10×2 | 1.92 |
| iCT-XL/2 | 675M | 1 | - | 34.24 | | MAR-H | 943M | 256×2 | 1.55 |
| Shortcut-XL/2 | 675M | 1 | 250 | 10.60 | | ***diffusion/flow*** | | | |
| MeanFlow-XL/2 | 676M | 1 | 240 | 3.43 | | ADM | 554M | 250×2 | 10.94 |
| ***1-NFE GANs from scratch*** | | | | | | LDM-4-G | 400M | 250×2 | 3.60 |
| StyleGAN-XL† | 166M | 1 | - | 2.30 | | SimDiff | 2B | 512×2 | 2.77 |
| BigGAN | 112M | 1 | - | 6.95 | | DiT-XL/2 | 675M | 250×2 | 2.27 |
| GigaGAN | 569M | 1 | 480 | 3.45 | | SiT-XL/2 | 675M | 250×2 | 2.06 |
| GAT-XL/2 | 602M | 1 | **60** | **2.18** | | SiT-XL/2+REPA | 675M | 250×2 | 1.42 |
| | | | | | | LightningDiT-XL/2 | 675M | 250×2 | 1.35 |

token dimensions, and sim is a similarity measure such as cosine similarity.

In short, the full discriminator and generator objectives are

$$\mathcal{L}_D = \mathcal{L}_D^{\text{adv}} + \lambda_{\text{aGP}}\mathcal{L}_{\text{aR1}} + \lambda_{\text{aGP}}\mathcal{L}_{\text{aR2}} + \lambda_{\text{REPA}}\mathcal{L}_{\text{REPA}}, \quad (11)$$

$$\mathcal{L}_G = \mathcal{L}_G^{\text{adv}}, \quad (12)$$

where $\lambda_{\text{aGP}}$ and $\lambda_{\text{REPA}}$ are the strength of gradient penalty and alignment objectives, respectively. For other details, we further elaborate them in Appendix.

## 3. Experiments

**Experimental settings.** We conduct class-conditional generation on ImageNet (Deng et al., 2009) at 256×256 and report FID (Heusel et al., 2017) 50K samples. We tokenize images using the pre-trained SD-VAE (Rombach et al., 2022) and train in the 32×32 latent space. We evaluate four model sizes (S/B/L/XL) following (Peebles & Xie, 2023) and use patch size $p=2$ unless noted.

We keep training hyperparameters fixed across scales except for the learning rate, which is scaled as in Sec. 2.4. We use a projection discriminator (Miyato & Koyama, 2018) and train all models for 50K iterations with batch size 512 (i.e., 20 epochs).

### 3.1. Comparison with prior arts

We compare the proposed method with various types of generative models, including one or two-step and multi-step GAN/diffusion/flow models. As reported in Tab. 1, our GAT-XL/2 achieves the state-of-the-art FID-50K on ImageNet-256, significantly enhancing the FID on 1-step generation (3.43 to 2.18). Notably, it reaches this performance with only 60 epochs, substantially fewer training

epochs than prior methods. This experimental result implies strong data efficiency of the proposed method and suggests further gains can be achieved with longer training. More importantly, it shows that GANs possess generative capabilities that are not significantly inferior to those of other generative models. We also provide qualitative comparison with one-step generative models, MeanFlow and StyleGAN-XL, in Appendix (Fig. 18)

### 3.2. Training GAT on various scales

**Model size.** We trained GAT across various model capacities, then measured the FID-50K for every 10K iterations. As shown in Fig. 3a, we observe that larger models consistently achieve lower FID, and this advantage mostly persists throughout training rather than appearing only at convergence. This scaling behavior shows that the training GAN can be easily scaled up, similar to other types of generative models, with minimal modification in hyperparameters.

**Patch size.** We further assess the robustness of the proposed method against tokenization granularity by performing experiments with a larger patch size of p=4 for the Small and Base configurations. As shown in Fig. 3b, the models are successfully trained and attain acceptable FID across patch sizes, indicating that the proposed method can be easily extended across various patch sizes.

**Inference and training cost (GFLOPs).** Model complexity is commonly measured by GFLOPs. Therefore, we also plot FID-50K against the transformer's computational cost measured in GFLOPs, and compute the correlation between the model's performance and its GFLOPs. As shown in Fig. 3c, we observe a strong negative correlation (-0.95): models with higher compute systematically yield better (lower) FID. Importantly, as in Fig. 3d, we observe

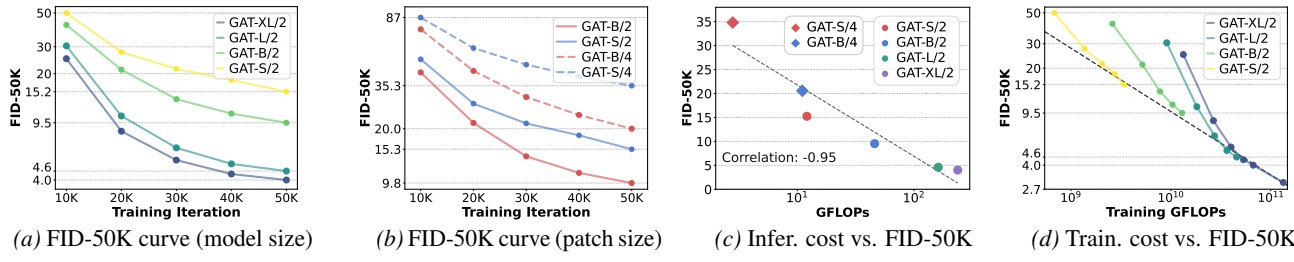

*(a) FID-50K curve (model size)*  *(b) FID-50K curve (patch size)*  *(c) Infer. cost vs. FID-50K*  *(d) Train. cost vs. FID-50K*

*Figure 3.* Scalability of GAT. (a) Training curve of FID-50K across the various model sizes shows that the performance is monotonically increasing as the model size is scaled up. (b) Training curve of FID-50K across the various patch sizes. With an identical number of parameters, we observe that the higher computational power of models enhances the generation capability. (c) We observe strong negative correlation between FID-50K and inference cost (GFLOPs), and (d) there is also a similar strong negative correlation between FID-50K and training cost (GFLOPs), proving that both higher compute of model and entire training systematically yields better (lower) FID. The dashed line in (d) shows a power-law fit, following $\text{FID}(C) \approx 3.52 \times 10^5 \cdot C^{-0.456}$, where $C$ is total training compute (GFLOPs).

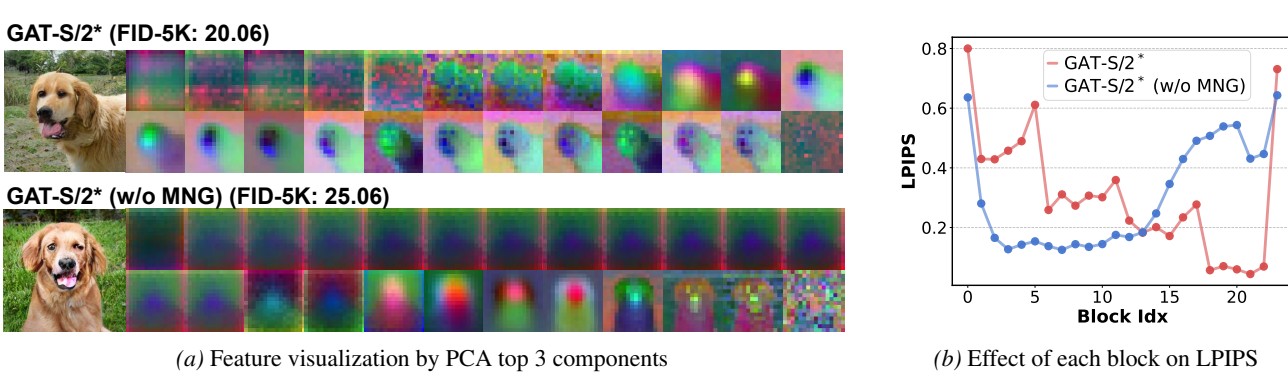

*(a) Feature visualization by PCA top 3 components*  *(b) Effect of each block on LPIPS*

*Figure 4.* Visualization of intermediate features of the generator and their effects on the generated images. (a) Both GAT models reveal the coarse-to-fine synthesis process, but without the Multi-level Noise-perturbed image Guidance (MNG), the generator's early layers become largely inactive, showing feature visualizations change only marginally, whereas our method activates these layers much earlier. (b) LPIPS distances while ablating Transformer blocks one by one. Without MNG, removing early blocks yields only minor changes in the output, despite those blocks producing coarse information, indicating computational inefficiency in the generator's early layers. GAT-S/2* doubles the number of blocks relative to GAT-S/2 for finer block-level analysis.

the same trend when considering total training compute (i.e., cumulative GFLOPs over the entire training run): FID-50K remains strongly negatively correlated with training compute, indicating that larger training budgets consistently improve generative performance. These results indicate that scaling improves performance and that the proposed GAT is scalable and effectively utilizes the scalable characteristics of transformer architectures. Note that inference costs are computed for a single forward pass of the generator. The dashed line in (d) shows a power-law fit, following $\text{FID}(C) \approx 3.52 \times 10^5 \cdot C^{-0.456}$, where $C$ is total training compute (GFLOPs). Details about Fig. 3d are provided in Appendix. A.15.

### 3.3. Early-layer inactivity analysis

As discussed earlier, we demonstrate that a vanilla GAT without MNG exhibits inactive features in early layers, which may significantly limit the benefits of scaling. To characterize this behavior, we visualize intermediate features for each transformer block using PCA. As shown in

Fig. 4a, early-layer features are highly redundant without MNG, suggesting that most early blocks remain inactive and weakly contribute to image synthesis. In contrast, our method yields well-distributed feature activations throughout the network.

Furthermore, we quantitatively measure the influence of each block on the generated image, as shown in Fig. 4b. In detail, we ablate each transformer block (i.e., replace it with an identity mapping), re-synthesize the image, and compute the LPIPS (Zhang et al., 2018) distance to the unablated output; smaller LPIPS values indicate a lower perceptual contribution. We compute these statistics on 10K images. Consistent with the above observation, the model without MNG shows weak early-layer contribution, with most of the generative process concentrated in later blocks. By contrast, our model exhibits a progressively decreasing contribution from early to late layers, aligning with MNG's intended coarse-to-fine behavior: intermediate layers receive sufficient guidance, responsibility is distributed across depth, and capacity is utilized more uniformly. Note that the last

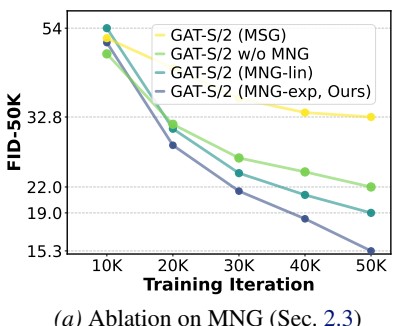
*(a) Ablation on MNG (Sec. 2.3)*

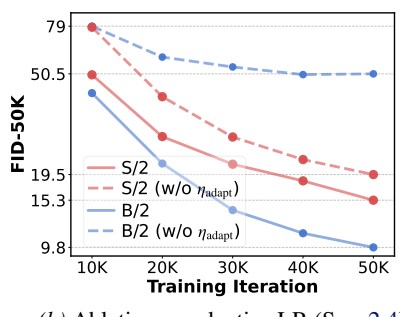
*(b) Ablation on adaptive LR (Sec. 2.4)*

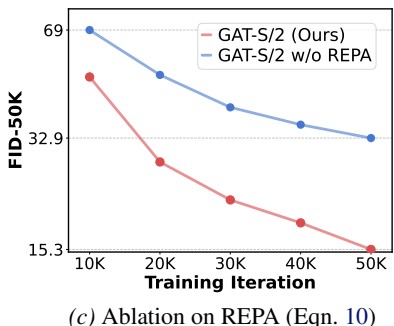
*(c) Ablation on REPA (Eqn. 10)*

*Figure 5.* Ablation study. (a) Multi-level Noise-perturbed image Guidance (MNG) consistently enhances the performance throughout the entire training (vs. w/o MNG) and also surpasses the original MSG-GAN, which degrades images by resize operation (vs. MSG). (b) Effect of adaptive learning rate scaling. Each model converges stably with its own $\eta_{adapt}$, while transferring it with another model's $\eta$ leads to severe degradation. (c) The REPA objective substantially improves performance, indicating that advances from diffusion models can transfer effectively to GAT.

layer tends to spike, likely because it is located immediately before the final synthesis output. We additionally compute (i) the similarity between each block's input and output and (ii) sensitivity to $z$, and find that MNG amplifies the functional contribution of individual blocks in both diagnostics, expected to enhance the benefits from scaling; detailed results are provided in Appendix A.7.

### 3.4. Ablation study

**Multi-level Noise-perturbed Guidance (MNG) (Sec 2.3).** We evaluate MNG quantitatively by comparing FID-50K training curves in Fig. 5a. We consider four variants: (i) MSG (replacing noising-based degradation with resize-based degradation, following MSG-GAN (Karnewar & Wang, 2020)), (ii) w/o MNG, (iii) MNG-lin (linear noise schedule), and (iv) MNG-exp (exponential noise schedule, our default setting). Across runs, our base setting, MNG-exp, consistently achieves the best (lowest) FID, outperforming both the no-MNG baseline and the linear schedule. Interestingly, MSG delivers the weakest performance. We hypothesize that, as reported in prior work (Lin et al., 2021; Kang et al., 2023), feeding the discriminator multi-scale outputs can overemphasize cross-scale consistency, which in turn suppresses generative quality. In contrast, our MNG perturbs a single degraded counterpart with stochastic noise at multiple levels, providing diversity without enforcing strict cross-scale alignment, and thereby avoiding the aforementioned failure mode.

**Adaptive learning rate (Sec. 2.4).** For each model, an appropriate learning rate is determined by the adaptive learning rate strategy (Fig. 5b), which ensures stable convergence. To assess the effectiveness of this strategy, we conduct a cross-check experiment by reusing configurations across scales (i.e., training GAT-S/2 with the $\eta_{adapt}$ of GAT-B/2; and vice versa). In this naive setting, reusing the configuration of another model, performance degrades substantially: GAT-S/2 converges slowly due to an overly small learning rate, while GAT-B/2 diverges under an excessively large learning rate. These results indicate that our adaptive learning rate strategy reliably selects a proper learning rate across scales without any manual tuning, a key factor for scalability.

**VFM alignment objective $\mathcal{L}_{REPA}$ (Eqn. 10).** We ablate the REPA objective, which aligns discriminator representations with those from a Vision Foundation Model (VFM), as in Fig. 5c. REPA significantly and consistently enhances the performance of the generator, although we impose a feature alignment objective only on the discriminator. This suggests that enhancing the discriminator's representation quality yields more informative adversarial feedback to the generator, leading to improved generation performance. Furthermore, this result implies that recent techniques developed for diffusion models using VFMs (Yao et al., 2025b; Chen et al., 2025) can transfer effectively to our GAT framework (e.g., GAT can leverage the performance of enhanced VAE (Leng et al., 2025), Tab. 9).

### 3.5. Further analysis and experiments

**Decoupled analysis of Generator and Discriminator scaling.** We analyze the relative contributions of $G$ and $D$ by scaling them individually. As shown in Fig. 6a, training remains stable and performance improves in both cases, but the gains from scaling the discriminator are notably larger.

This suggests that, because the generator only learns through the discriminator's feedback, overall performance is effectively bottlenecked by how well the discriminator models the data distribution and shapes the real–fake decision boundary, so scaling up the discriminator, thereby providing sharper and more informative gradients, yields larger gains than merely increasing the generator's capacity. In addition, this observation aligns with our discussion on the importance of representation learning in discriminators,

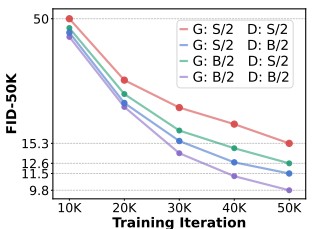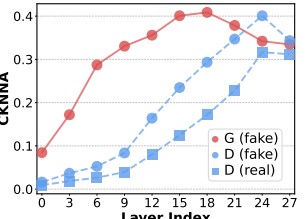

(a) Decoupled scaling of G and D    (b) CKNNA of G and D

*Figure 6.* Further analysis. (a) Scaling G and D separately shows both impact FID, while scaling D is more effective than G. (b) Feature alignment against DINOv2-g measured by CKNNA using real and fake data. We observe that the features obtained from fake data show higher alignment with VFM than real data.

*Table 2.* Experiments on ImageNet-512. Training at high-resolution (512) achieves a similar FID-50K with fewer epochs.

| Resolution | Model | Epochs | FID-50K |
|---|---|---|---|
| 256×256 | GAT-XL/2 | 20 | 4.02 |
| 512×512 | GAT-XL/2 | 15 | 4.04 |

highlighting its central role in adversarial learning.

**Representation Alignment of Generator and Discriminator.** Recent work on diffusion models (Yu et al., 2025) shows that generation quality tends to be proportional to the degree of feature alignment to Vision Foundation Models (VFMs). Motivated by this, we evaluate the feature-alignment metric CKNNA (Huh et al., 2024) of both the generator and discriminator against DINOv2-g on real and fake data (Fig. 6), and observe fake data report high CKNNA compared to real data. Our intuition is that generated samples tend to fall within the discriminator's well-established feature space, where representations are most reliable. In this space, the discriminator can provide strong and effective guidance, from which the generator consistently benefits, leading to higher-quality synthesis. Accordingly, as the generative performance of $G$ is tightly coupled with the representation quality of $D$, further strengthening discriminator representations may be a promising direction for future work.

**Experiments on high-resolution data (ImageNet-512)** We verify that our model scales favorably to higher resolutions by training GAT-XL/2 on ImageNet at 512×512 resolution (Tab. 2). We train for 15 epochs and obtain an FID-50K of 4.04, which is comparable to the 20 epochs result at 256×256 resolution. This suggests that our framework can achieve similarly strong performance with fewer epochs, even at higher resolutions. Qualitative samples at 512×512 are provided in Fig. 7.

**Additional evaluation beyond Inception-FID.** We further evaluate GAT using CLIP-FID (Kynkäänniemi et al., 2023b) and diversity-related metrics to verify that the improvement is not specific to Inception-based FID. This is particularly

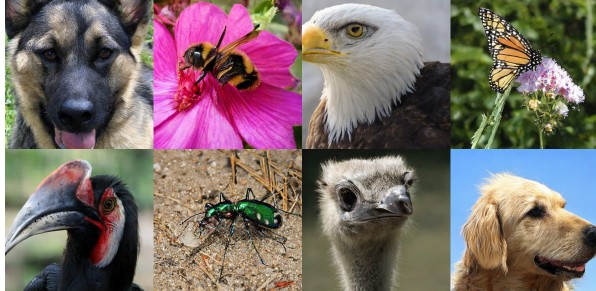

*Figure 7.* Examples from GAT-XL/2 on ImageNet-512.

*Table 3.* Additional comparison of one-step generators

| Method | CLIP-FID↓ | FID↓ | Prec.↑ | Rec.↑ |
|---|---|---|---|---|
| MeanFlow-XL/2 | 3.17 | 3.41 | 0.810 | 0.525 |
| StyleGAN-XL | 2.62 | 2.30 | 0.780 | 0.530 |
| GAT-XL/2 (40 epo) | 2.21 | 2.96 | 0.805 | 0.527 |
| GAT-XL/2 (60 epo) | **1.86** | **2.18** | 0.796 | **0.572** |

relevant because ImageNet-pretrained discriminator backbones can partially align with the Inception feature space used by FID. As shown in Tab. 3, GAT-XL/2 achieves the best CLIP-FID among the compared one-step models. The 60-epoch GAT-XL/2 improves CLIP-FID from 2.62 to 1.86 compared to StyleGAN-XL, while also achieving higher Recall. This suggests that GAT improves generation quality in a complementary feature space and preserves stronger sample diversity.

**Compatibility with large-batch training.** We also test whether the width-aware learning-rate rule composes with standard batch-size scaling. For this experiment, we train GAT-S/2 with a $4\times$ larger batch size and adjust the learning rate according to the square-root scaling rule for adaptive optimizers (Malladi et al., 2022). This batch-size scaling rule addresses a different axis from our width-aware rule: it compensates for the reduced gradient stochasticity induced by larger minibatches, whereas our rule keeps the functional update scale approximately consistent across model widths. As shown in Fig. 8, under this combined setting, GAT-S/2 reaches comparable FID-50K to the default setting, 15.86 versus 15.27, while requiring only one quarter of the training iterations. This result suggests that our scaling recipe is not a competing alternative to existing large-batch heuristics, but can be naturally composed with them to support more practical high-throughput training regimes.

**Synthesis examples from semantic latent manipulation and multi-level outputs.** We further present latent interpolation results (Fig. 10), highlighting a characteristic behavior of GAN-style generative models: smoothly traversing the latent space yields natural, continuous transitions between images. Importantly, observing such smooth transitions in our GAT model supports that GANs can retain a coherent

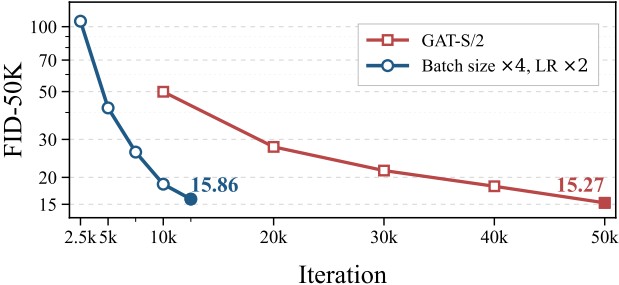

*Figure 8.* **Compatibility with large-batch training.** With a $4\times$ larger batch size and square-root learning-rate scaling on top of our width-aware rule, GAT-S/2 reaches comparable FID-50K to the default setting in one quarter of the training iterations.

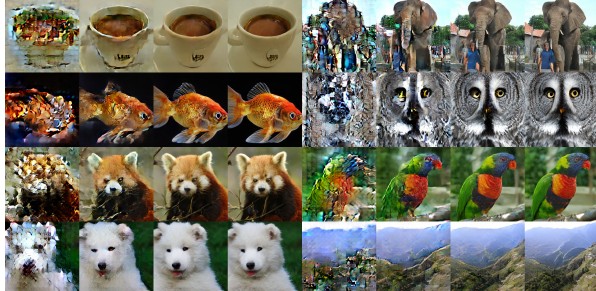

*Figure 9.* Examples of stage-wise outputs produced by MNG.

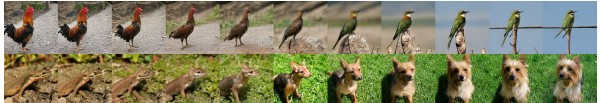

*Figure 10.* Latent interpolation examples on ImageNet-256.

and well-structured latent space even when scaled up, indicating stable training and effective representation learning at scale. In addition, we provide visual examples of the stage-wise multi-level outputs produced by the generator (Fig. 9). These examples show how intermediate outputs progressively evolve toward the final synthesis result, where earlier stages capture coarse semantic structure and later stages refine fine-grained appearance details. This behavior is consistent with the design of MNG, which assigns stronger perturbation to earlier outputs and weaker perturbation to later outputs, thereby encouraging a coarse-to-fine synthesis trajectory without introducing an explicit resolution hierarchy in the transformer backbone. Additional examples of unsupervised and text-driven image editing are provided in the Appendix (Fig. 16, Fig. 17).

## 4. Related Works

**Generative Adversarial Networks (GANs)** are trained via an adversarial game between a generator and a discriminator, with progress driven by architectural and objective-level innovations. Convolutional GANs, notably the StyleGAN family (Karras et al., 2019; 2020), have achieved strong results and have been extended to large-scale text-to-image generation (Kang et al., 2023; Sauer et al., 2023), but remain largely tied to pixel-space synthesis. Transformer-based GANs have also been explored (Jiang et al., 2021; Zhao et al., 2021; Lee et al., 2022b); however, their reliance on substantial departures from plain transformer designs and careful hyperparameter tuning often hinders scalability. On the objective side, a range of adversarial losses (Goodfellow et al., 2014; Arjovsky et al., 2017; Lim & Ye, 2017; Mao et al., 2017) and regularizers (Mescheder et al., 2018; Gulrajani et al., 2017) have been proposed, with recent work improving stability via relativistic objectives and gradient-based regularization (Huang et al., 2024; Bhaskara et al., 2022; Wu et al., 2021). In this work, we establish a fully transformer-based GAN framework in the latent space of a VAE and empirically study its scalability.

**Scalability of generative models** has been central to recent advances. Diffusion and flow models show consistent gains from transformer backbones (Peebles & Xie, 2023; Ma et al., 2024) and systematic scaling with data and compute (Liang et al., 2026), further enabled by latent tokenizers (Rombach et al., 2022; Yao et al., 2025b; Chen et al., 2025) for efficient high-resolution synthesis (Esser et al., 2024b; Podell et al., 2024). Autoregressive models similarly benefit from transformer scaling across class-conditional generation and text-to-image synthesis (Chang et al., 2022; Tian et al., 2024; Han et al., 2025). Motivated by these trends, we revisit GANs through transformer-based latent architectures, aiming to preserve single-step inference while inheriting the favorable scaling behavior of transformers.

## 5. Conclusion

We revisit GAN scalability by pairing VAE-latent training with plain transformer generators and discriminators. Addressing early-layer underuse and scale-coupled instability with lightweight intermediate supervision and width-aware learning-rate scaling yields GAT, which trains reliably from S to XL and reaches state-of-the-art one-step ImageNet-256 in 60 epochs ($4\times$ fewer than strong baselines). We hope our work will be a strong step toward scaling GANs.

## Impact statement

This work improves the efficiency and quality of transformer-based image generation, which can benefit creative applications and data augmentation. However, stronger and cheaper generative models may also be misused to produce deceptive synthetic media (e.g., deepfakes and misinformation). Responsible release and deployment practices (e.g., provenance/watermarking and access controls) are recommended to mitigate misuse.

## Acknowledgements

This work was supported in part by MSIT/IITP (No. RS-2022-II220680, RS-2020-II201821, RS-2019-II190421, RS-2024-00459618, RS-2024-00360227, RS-2024-00437633, RS-2024-00437102, RS-2025-25442569), MSIT/NRF (No. RS-2024-00357729), KNPA/KIPoT (No. RS-2025-25393280), and SEMES-SKKU collaboration funded by SEMES.

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

# A. Appendix

## A.1. Implementation details

We provide the configurations for all model sizes, including the parameter counts of the generator and discriminator in Tab. 4. Also, we report the detailed FID-50K score at 50K iterations in Tab. 5, which is used for visualizing Fig. 3c.

**Generator**  We design our models following common conventions from ViT (Dosovitskiy et al.) and StyleGAN (Karras et al., 2020). We use a latent code $z$ of dimension $d_z = 64$, and initialize the class embedding with the standard ViT token scale of 0.02. The mapping network is a shallow MLP whose width matches the transformer hidden dimension; it consists of two linear layers with a single nonlinearity, using SiLU in line with transformer practice.

Following StyleGAN, we train the mapping network with a learning rate that is 100× smaller than the rest of the generator. The main GAT block is as described in the paper, and we additionally adopt techniques reported to improve transformer performance, Rotary Positional Embeddings (RoPE) (Su et al., 2024), SwiGLU-FFN (Shazeer, 2020), and qk-normalization. Finally, all scaling parameters produced from style codes are initialized to have a variance 0.1.

For the number of intermediate outputs $K$, we use $k = 4$ for every model size. These outputs are synthesized at uniform intervals across the generator's GAT blocks. For example, in GAT-XL/2 with 28 layers, we take an output every 7 layers.

**Discriminator**  The discriminator largely follows a standard ViT, with the sole exception that each module output is gated by a Layerscale factor; all Layerscale vectors are initialized to 0.1. Similar to the generator, every transformer block uses RoPE, a SwiGLU feed-forward network, and qk-normalization. The projection layer, for the VFM-alignment objective, $P$ follows REPA (Yu et al., 2025) and is implemented as a 3-layer MLP with a hidden dimension of 2048. Also, we deploy DINOv2-B as a vision foundation model to align with.

During training, we apply differentiable augmentation (Zhao et al., 2020). To combine it with the noise-adding operations (approximated GP and multi-level noise-perturbation guidance), we proceed as follows: upon receiving an input image, we first add the perturbation used for the approximated GP, then apply the augmentation, and finally apply the multi-level noise perturbations. For the approximated GP, the same noise magnitude is used for all noise levels ($\sigma = 0.01$).

**Noise sampling and schedule for MNG**  We design the image signal doubles at each successive output. Since the final output should be a clean image, for $k = 4$ we set

$$(\alpha_1, \alpha_2, \alpha_3, \alpha_4) = (0.125,\ 0.25,\ 0.5,\ 1.0).$$

In addition, we build the noise at each level cumulatively, starting from the noise added to the clean image and accumulating the newly sampled noise for constructing lower-level noise.

Given noise $\epsilon_k$ at level $k$, we obtain the noise $\epsilon_{k-1}$ at level $k-1$ as follows:

$$\epsilon_{k-1}\ =\ r_k\,\epsilon_k\ +\ \sigma_k\,\eta_k, \qquad \eta_k \sim \mathcal{N}(0, I),$$

where the signal schedule is $\alpha_1 < \cdots < \alpha_K$ with $\alpha_K = 1$, and

$$r_k\ =\ \frac{\alpha_{k-1}}{\alpha_k}, \qquad \sigma_k = \sqrt{1 - r_k^2}.$$

This noise sampling preserves the variance of $\epsilon_k$ at every level while keeping the noise already sampled at the higher levels.

**Other hyperparameter**  Basically, every hyperparameter is shared across any size of models, except the learning rate. We train with a gradient-penalty coefficient $\lambda_{\text{aGP}} = 1 \times 10^{-1}$ and VFM alignment objective coefficient $\lambda_{\text{REPA}} = 1$. We use a fixed gradient-penalty coefficient for the first 40 epochs on ImageNet-256, and set it to 10 for epochs 40–60. The optimizer is AdamW with $(\beta_1, \beta_2) = (0.0,\ 0.99)$ (following common GAN practice such as StyleGAN). We apply the exponential moving average (EMA) to the generator with decay 0.999. Also, we use a batch size of 512, `bfloat16` precision, gradient checkpointing, and PyTorch Scaled Dot-Product Attention (SDPA) implementation.

For learning rate, we use $4 \times 10^{-4}$ as the *base* learning rate for the GAT-S model. After applying our adaptive learning rate rule, the per-size learning rates are: (GAT-S, GAT-B, GAT-L, GAT-XL) = $(4 \times 10^{-4},\ 2 \times 10^{-4},\ 1.5 \times 10^{-4},\ 1.33 \times 10^{-4})$.

*Table 4.* Model configuration and parameter counts (M = million).

| Model | Layers | Dim | Heads | G params | D params |
|-------|--------|-----|-------|----------|----------|
| GAT-S | 12 | 384 | 6 | 29.36M | 39.21M |
| GAT-B | 12 | 768 | 12 | 116.75M | 104.68M |
| GAT-L | 24 | 1024 | 16 | 408.75M | 323.04M |
| GAT-XL | 28 | 1152 | 16 | 602.25M | 467.68M |

*Table 5.* FID at 50K iter. across model sizes.

| Model | FID-50K |
|-------|---------|
| GAT-XL/2 | 4.021 |
| GAT-L/2 | 4.600 |
| GAT-B/2 | 9.534 |
| GAT-S/2 | 15.237 |

*Table 6.* Ablation on MNG (FID-5K).

| Model | FID-5K |
|-------|--------|
| GAT-S/2 ($lr$=4e-4) | 22.080 |
| GAT-B/2 ($lr$=2e-4) | 15.610 |
| GAT-S/2 ($lr$=2e-4, w/o $\eta_{adapt}$) | 26.410 |
| GAT-B/2 ($lr$=4e-4, w/o $\eta_{adapt}$) | 56.720 |

## A.2. Additional related works

We simply explain the baselines that we compare with in Tab. 1.

- **VQGAN** (Esser et al., 2021) introduce the GPT-like autoregressive model on the discretized visual tokens to build the generative model.

- **ADM** (Dhariwal & Nichol, 2021) proposes the U-Net-based diffusion architecture with a classifier guidance, firstly beating the GAN counterpart in image generation task.

- **MaskGIT** (Chang et al., 2022) proposes a parallelized decoding strategy to improve the inference speed of autoregressive models.

- **LDM** (Rombach et al., 2022) proposes to train diffusion model on the latent space of pre-trained VAE, enhancing the generation capability and inference speed.

- **SimDiff** (Hoogeboom et al., 2023) improves the standard denoising diffusion model to train directly in pixel space on high-resolution images.

- **DiT** (Peebles & Xie, 2023) proposes replacing the conventional U-Net backbone in diffusion models with plain (non-hierarchical) transformers with AdaLN-zero layer.

- **iCT** (Song & Dhariwal, 2024) introduces distillation-free consistency training recipe, which surpasses previous consistency distillation.

- **SiT** (Ma et al., 2024) conducts an in-depth study showing that transitioning from discrete diffusion to continuous flow matching makes DiT training more efficient.

- **VAR** (Tian et al., 2024) introduces visual autoregressive model that substitutes spatial autoregression with progression across scales.

- **MAR** (Li et al., 2024) proposes a framework for training autoregressive models on continuous tokens by introducing a shallow diffusion model to sample the next token.

- **Shortcut** (Frans et al., 2025) learns the shortcut between two apart timestep to predict the single-step denoising direction.

- **iMM** (Zhou et al., 2025) proposes the method to train few-step generators from scratch by using self-consistent interpolants and matching all moments along the data.

- **MeanFlow** (Geng et al., 2026) introduces one-step generative framework which predicts average velocity, the time integral of the instantaneous velocity.

- **STARFlow** (Gu et al., 2026) proposes latent normalizing flows that scale to high-resolution synthesis and billion-parameter regimes.

- **Lightning-DiT** (Yao et al., 2025a) proposes a VAE trained to yield a generation-friendly latent space via distillation from a vision foundation model.

### A.3. Epoch-to-FID comparison

Fig. 11 compares training efficiency via Epoch-to-FID on ImageNet 256×256. Among single-step generators (i.e., 1-NFE models), our method (GAT) achieves strong FID with substantially fewer training epochs than prior approaches with reported epoch counts. In particular, GAT-XL/2 reaches low FID within 60 epochs, while representative baselines require several hundred epochs to attain comparable quality (e.g., Shortcut-XL/2 at 250 epochs and GigaGAN at 480 epochs). Overall, the results suggest that GAT improves data and training efficiency, reducing the amount of training needed to obtain high-quality samples.

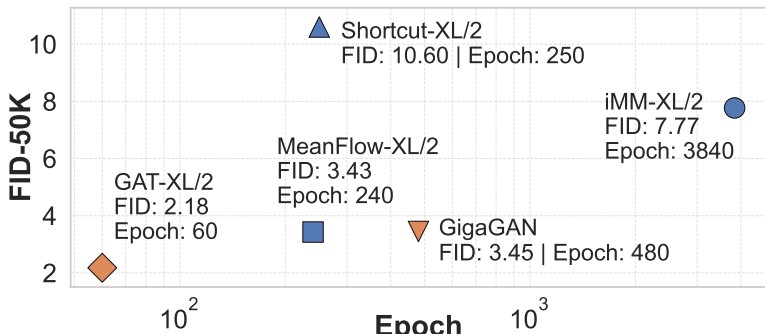

*Figure 11.* **Epoch-to-FID comparison on ImageNet 256×256 (FID-50K).** Our method (GAT) achieves lower FID with fewer training epochs than prior approaches, indicating improved data and training efficiency.

### A.4. Exact FID values for Fig. 3c and Fig. 5a

Fig. 3c and Fig. 5a present the GFlops vs. FID-50K comparison and the ablation results on MNG, respectively. To improve clarity, we provide the exact FID scores corresponding to these plots. Specifically, Tab. 5 lists the FID values used in Fig. 3c, and Tab. 6 reports the FID results for the MNG ablation in Fig. 5a.

### A.5. Training curve of GAT-XL/2 (FID-50K)

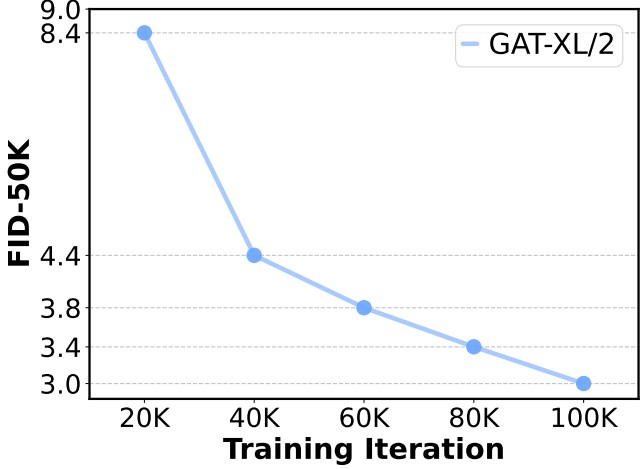

*Figure 12.* Training curve of GAT-XL/2 until 40 epochs.

We additionally report the FID-50K training curve for GAT-XL/2 up to 100K iterations (i.e., 40 epochs). The metric decreases monotonically, suggesting that further training would likely yield additional improvements.

## A.6. Training time and Inference cost of GAT

**Training time**  Because GAT updates both the generator and discriminator at every iteration, a single training step is expensive than that of a pure diffusion model. In our measurements, processing the same number of images takes roughly $5\times$ longer per iteration compared to a DiT-style diffusion transformer. However, this does *not* imply a longer overall time-to-quality. For example, DiT-XL/2 requires about 1400 epochs on ImageNet-256 to reach FID 2.27, which is reported to take roughly 36 days on 8×A100 GPUs (Zheng et al., 2024). In contrast, our GAT-XL/2 attains a better FID-50K of 2.18 after only 60 epochs, corresponding to approximately 18 days on a less powerful setup with 8×RTX A6000 GPUs. In addition to the lower wall-clock training time, this also means GAT observes fewer total passes over the data, indicating better data efficiency. Importantly, the modern one-step diffusion model, such as MeanFlow (Geng et al., 2026), also requires additional computation of the gradient, so it also requires additional time.

Furthermore, most of this training cost is dominated by the discriminator, which must repeatedly distinguish real from fake samples and evaluate the gradient-penalty term. We therefore expect that the pretrained feature extractor as a discriminator, in the spirit of projected GANs (Sauer et al., 2021), and then fine-tuning only the later layers for real/fake discrimination could substantially reduce this cost, as follows previous GANs (Sauer et al., 2022; 2023). Exploring such pretrained discriminator and its impact on training time and stability is an interesting direction for future work.

*Table 7.* Inference latency and memory footprint when generating 64 ImageNet-256 samples on a single Titan RTX GPU.

| Model | NFE | Time / image (s) | Peak memory (MB) |
|---|---|---|---|
| DiT-XL/2 | 250 | 15.2612 | 4525.85 |
| GAT-XL/2 (ours) | 1 | 0.0773 | 6028.11 |

**Inference cost.**  GAT shares almost the same pure transformer backbone as DiT (Peebles & Xie, 2023), since we deliberately keep the architecture close to a plain transformer block. Consequently, the *per-step* inference cost and memory footprint for a single NFE are very similar between GAT and DiT at matched width and depth. The key difference comes from the number of function evaluations (NFE): while DiT typically requires around 250 denoising steps, GAT is a single-step generator (1 NFE). In practice, this translates into roughly two orders of magnitude speedup; in our measurements (Table. 7), GAT-XL/2 is about $200\times$ faster than DiT-XL/2 at comparable quality.

Furthermore, most of the end-to-end inference memory in both models is dominated by the transformer backbone. Since GAT performs truncation and guidance directly in the latent space before the backbone, it does not require additional passes through the heavy network, so the backbone-side memory usage remains essentially comparable to DiT. Any residual difference in peak VRAM mainly comes from lightweight auxiliary heads (e.g., MNG feature caching), rather than from the core architecture.

Concretely, we measure inference latency and memory on a single Titan RTX GPU when generating 64 ImageNet-256 samples, as shown in Table. 7. For DiT-XL/2, using the standard 250-step sampler, generating one image takes 15.2612 seconds with a peak memory usage of 4.5 GB. In contrast, GAT-XL/2 with 1-NFE generation requires only 0.0773 seconds per image, corresponding to roughly a $200\times$ speedup, with a peak memory usage of 6.0 GB. The slightly higher VRAM footprint for GAT is mainly due to caching intermediate features for the MNG guidance head; the transformer backbone itself has a comparable memory cost to DiT at similar width and depth.

## A.7. Further Motivation for Multi-level Noise-perturbed Guidance

**Motivation**  The central motivation for MNG comes from how other generative models couple their inputs with the final generated image that takes the supervision. In diffusion and autoregressive models, the input and output at each step are tightly linked by the training objective: step-wise denoising in diffusion, or next-token prediction in autoregressive models. This step-wise supervision explicitly enforces a strong relationship between intermediate step inputs and clean targets throughout the depth of the network, so that intermediate representations must remain informative with respect to the final sample.

In contrast, a standard GAN generator typically receives a low-dimensional latent code that is mapped to a constant input, while the adversarial loss is applied only to the final image via the discriminator. As long as the last few blocks can synthesize images that fool the discriminator, there is little incentive for earlier blocks to maintain rich structure or to remain

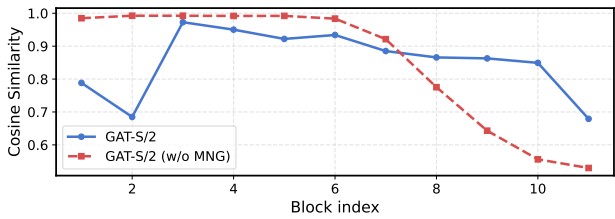

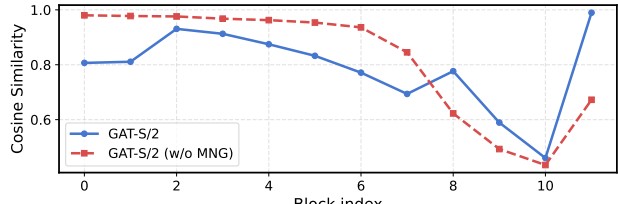

*(a)* Feature similarity between block input and output.    *(b)* Feature similarity across different input noises $z$.

*Figure 13.* Quantitative analysis of early-layer inactivity with and without MNG. (a) Cosine similarity between the input and output features of each block, averaged over all spatial locations, tokens, and images. (b) Cosine similarity between features obtained from different input noises $z$ for the same class. Without MNG, both similarities are close to 1 in the early blocks, indicating that they induce only minor feature changes and are weakly influenced by the input noise. With MNG, the similarities in early blocks are reduced for both measures, showing that MNG increases early-layer update magnitude and noise responsiveness.

"The train is on the tracks by the road."  "A small kitchen with low a ceiling"  "A girl on skis with a snowy background"  "A clear vase with flowers sitting a little table"  "Baked pizza with herbs displayed on serving tray at table"  "On a baseball diamond a player holds a bat while in front of a catcher in gray"

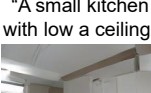 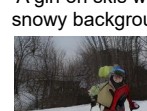 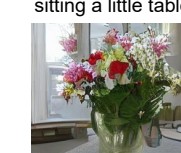 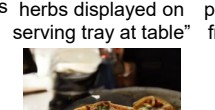 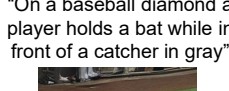 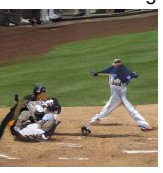

*Figure 14.* Examples of text-to-image generation on MS-COCO-256.

sensitive to the latent code. Optimization can therefore push most of the representational burden toward the blocks closest to the output, leaving early layers underutilized or close to constant mappings, as observed in our empirical analysis.

The proposed multi-level noise-perturbed guidance is designed to counteract this asymmetry. By injecting noise signals at multiple depths and supervising the corresponding intermediate outputs with different noise strengths, the generator is encouraged to respond meaningfully to these perturbations across its entire depth rather than relying predominantly on the final blocks. This induces a coarse-to-fine usage of layers: earlier stages are trained to capture global, noise-robust structure under strong perturbations, while later stages progressively refine fine details as the noise level decreases. Consequently, the generator is nudged away from degenerate solutions where early layers collapse, and toward a regime where representational responsibility is more evenly distributed across layers, which we show leads to improved utilization and stronger overall performance.

**Additional analysis on inactivity of early layers** To quantify the inactivity of early layers, we performed a block-ablation study, as summarized in Fig. 4b. For each transformer block, we remove the block, regenerate the corresponding images, and measure the resulting change using a perceptual distance metric. We observe that, without MNG, ablating early blocks causes only minor changes in the generated images, even though these layers are expected to encode global, structural information. This suggests that the early part of the network is underutilized.

We further analyze this phenomenon by measuring feature similarity during generation. To this end, we sample 64 images per class for all classes using GAT-S/2. Then, we compute (1) the feature similarity between the input and output of each block, and (2) the similarity across different input noises $z$ (Fig. 13a and Fig. 13b). Similarity is defined as the cosine similarity at corresponding spatial locations, averaged over all tokens and images. For the model without MNG, both similarities are very high in the early layers (often close to 1), indicating that these blocks (i) induce only small changes in the features and (ii) are weakly influenced by the input noise. This is consistent with early-layer inactivity. In contrast, with MNG the similarities in early blocks are noticeably reduced for both measures, indicating that MNG increases the amount of feature change in early layers and makes them more responsive to the input noise.

### A.8. Experiments on Text-to-Image generation (MS-COCO)

We further evaluate our framework on text-to-image generation using MS-COCO (Lin et al., 2014) at $256^2$ resolution. Following the U-ViT (Bao et al., 2023) setting (Bao et al., 2023), we train on the MS-COCO training split and report FID on

*Table 8.* Text-to-image generation on MS-COCO at 256 resolution. We report FID and the number of function evaluations (NFE) at sampling time (lower is better). Methods marked with $^*$ use a CLIP image encoder.

| Method | Type | NFE | FID |
|---|---|---|---|
| Frido (Fan et al., 2023) | Diffusion | 200 | 8.97 |
| VQ-Diffusion (Gu et al., 2022) | Discrete diffusion | 100 | 19.75 |
| U-Net (Bao et al., 2023) | Diffusion | 50 | 7.32 |
| U-ViT-S/2 (Bao et al., 2023) | Diffusion | 50 | 5.95 |
| U-ViT-S/2 (Deep) (Bao et al., 2023) | Diffusion | 50 | 5.4 |
| AttnGAN (Xu et al., 2018) | GAN | 1 | 35.49 |
| DM-GAN (Zhu et al., 2019) | GAN | 1 | 32.64 |
| DF-GAN (Tao et al., 2022) | GAN | 1 | 19.32 |
| XMC-GAN (Zhang et al., 2021) | GAN | 1 | 9.33 |
| LAFITE$^*$ (Zhou et al., 2022) | GAN | 1 | 8.12 |
| MM-GAT (Ours) | GAN | 1 | 7.98 |

the validation set. For text encoding, we use a frozen CLIP text encoder, and adopt an MM-DiT (Esser et al., 2024a)-style conditioning scheme where the generator additionally receives the CLIP (Radford et al., 2021) word tokens as input (we refer to this model as MM-GAT), while the CLIP sentence embedding (i.e., the [eot] token) is used as a global conditioning signal. We set a hidden dimension of 768 and a model depth of 24.

As shown in Table 8, MM-GAT attains competitive performance: although it slightly underperforms the best U-ViT variants in FID, it outperforms prior GAN-based approaches with 1-NFE. In particular, MM-GAT achieves a lower FID than LAFITE (Zhou et al., 2022), despite not using a CLIP image encoder. We emphasize that this is a deliberately lightweight, first-pass extension of GAT to the text-conditional setting, and we expect that modest additional tuning of this design could further improve performance. We also show the generated examples in Fig. 14.

## A.9. Experiments on unconditional generation (FFHQ-256)

Our main experiments on ImageNet were designed to study how a transformer-based GAN scales in a complex, diverse, large-scale setting. To additionally verify the feasibility of unconditional training, we also train GAT-B/2 on FFHQ-256 (Fig. 15) for 25M images. When compared against a DiT-B/2 baseline trained under the identical size of generator and the same amount of data, our model achieves an FID of 9.74 versus 10.49 for DiT-B/2, indicating that GAT can successfully support unconditional generation and attains competitive performance with diffusion transformers even in unconditional generation.

## A.10. Intuition Behind the Width-Aware Learning Rate Rule

Eqn. 5 defines our width-aware learning rate schedule $\eta(C)$, which is designed to keep the *functional* update of the network approximately invariant as we change the channel dimension $C$. Here we provide additional intuition for this choice without a full mathematical proof.

Throughout the paper, when we refer to the *speed of change* of a network $f_\theta$, we mean the typical change in its outputs after a single optimization step, for example

$$\mathbb{E}_x\big[\|f_{\theta_{t+1}}(x) - f_{\theta_t}(x)\|\big], \tag{13}$$

where the expectation is over training samples $x$. Intuitively, this quantity measures how aggressively the function implemented by the network is updated per step, as opposed to the raw magnitude of parameter updates.

We make the following simplifying assumption, which is standard in analyses of wide neural networks: a hidden vector can be modeled as

$$x \in \mathbb{R}^C, \qquad x_i \sim \mathcal{N}(0, 1) \text{ i.i.d.,} \tag{14}$$

where $C$ is the channel dimension and we treat the channels as approximately independent and unit-variance. In this regime,

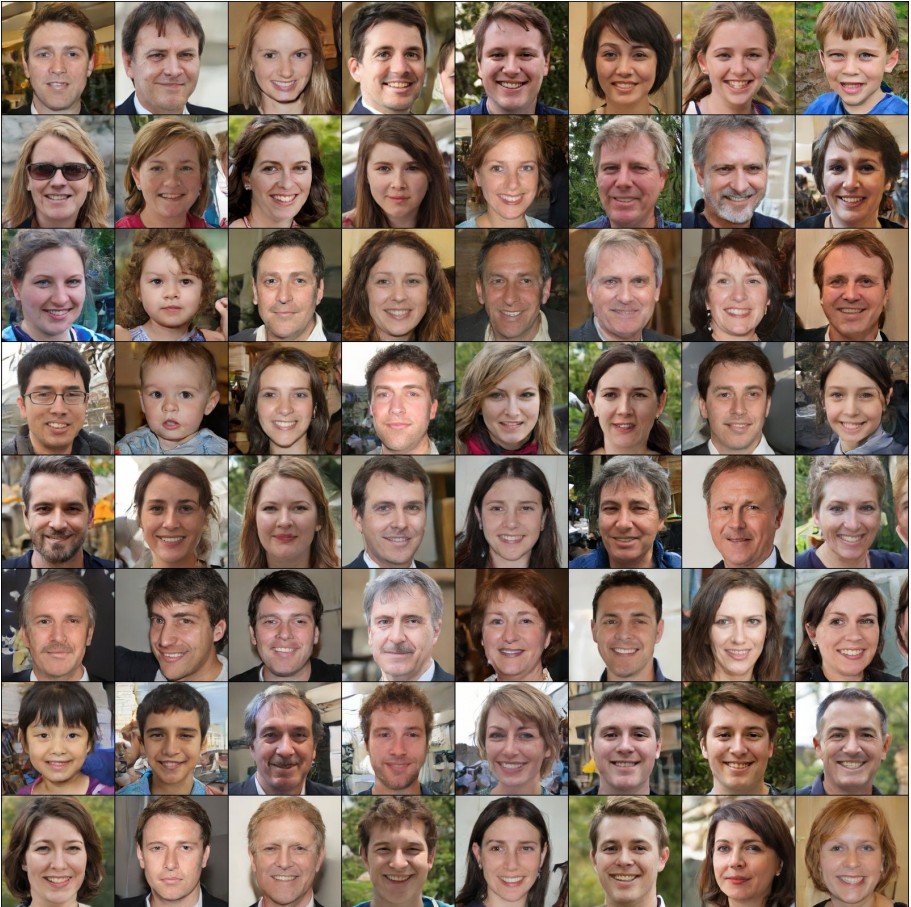

*Figure 15.* Unconditional generation examples from GAT-B/2 on FFHQ-256.

the squared norm of the activation vector satisfies

$$\|x\|^2 = \sum_{i=1}^{C} x_i^2 \approx C, \tag{15}$$

so the "energy" of a feature vector grows approximately linearly with width.

For intuition, consider a single linear layer with scalar output

$$f_\theta(x) = w^\top x. \tag{16}$$

If the loss gradient with respect to this scalar output is $g$, and the learning rate is $\eta$, then a single SGD step updates the weights as

$$w' = w - \eta g x. \tag{17}$$

Evaluating the updated layer on the *same* input $x$, the output changes by

$$f_{\theta'}(x) - f_\theta(x) = (w')^\top x - w^\top x \tag{18}$$

$$= (w - \eta g x)^\top x - w^\top x \tag{19}$$

$$= -\eta g \|x\|^2. \tag{20}$$

Under the standardized-channel assumption $\|x\|^2 \approx C$, the typical magnitude of this per-step output change scales roughly as

$$|f_{\theta'}(x) - f_\theta(x)| \propto \eta C. \tag{21}$$

In words, for a *fixed* learning rate $\eta$, wider networks (larger $C$) tend to change their outputs more per step, simply because their activations (and hence their effective updates) carry more energy. The same mechanism applies layer by layer, so the end-to-end change in $f_\theta(x)$ inherits a similar dependence on $C$.

To keep the functional update scale approximately stable when we vary the width, we therefore choose a learning rate that decreases inversely with the channel dimension, as in Eq. (4),

$$\eta(C) \propto \frac{1}{C}, \tag{22}$$

so that the product $\eta C$ remains roughly constant. This, in turn, stabilizes the per-step change of intermediate features and final outputs as we transition from smaller to larger GAN architectures.

### A.11. Relation to Depth and Batch-Size Scaling

Our width-aware learning-rate rule in Eqn. 5 is derived from the goal of keeping the per-step change in the network outputs approximately constant. This principle is not inherent to width alone and, in principle, can be extended into a more comprehensive scaling law that also accounts for depth and batch size.

For deep transformer-style residual networks, stacking $L$ blocks increases the cumulative effect of each update. Under the same "constant functional update" view, one could combine our width-based rule with an additional depth-dependent factor, e.g., a $\sqrt{L}^{-1}$-type correction, or equivalently, adjust Layerscale initialization as a function of $L$ so that the overall update magnitude of the network remains similar across depths. Along the batch axis, our rule can be composed with standard learning-rate scaling heuristics used for large-batch transformer training, such as linear scaling ($\eta \propto B$) or square-root scaling ($\eta \propto \sqrt{B}$), where $B$ denotes the batch size.

In this work, however, we intentionally focus on width for both conceptual clarity and empirical support. We conducted preliminary experiments in which we modified Layerscale initialization to compensate for depth, but within the depth range considered in our GAN architectures, this did not yield noticeable gains in stability or performance. Moreover, all of our main experiments use a fixed batch size of 512, so we do not yet have systematic evidence to justify incorporating batch-size dependence directly into the rule. Extending our width-aware learning-rate schedule to jointly cover width, depth, and batch size remains a promising direction for future work.

### A.12. Image Editing by manipulating the latent space of GAT

To assess the transferability and robustness of the learned GAT latent space beyond unconditional sampling, we additionally evaluate the compatibility of the learned latent space with off-the-shelf editing methods. In particular, we apply GANSpace (Härkönen et al., 2020), which discovers unsupervised editing directions and manipulates the generation process along them, and StyleCLIP (Patashnik et al., 2021), which steers generated images to match a given text prompt. As illustrated in Fig. 16 and 17, both editing techniques transfer cleanly to GAT, producing smooth and semantically meaningful variations, indicating that the learned latent space supports robust, reusable controls rather than overfitting to a single generative task.

### A.13. Qualitative comparison with other methods

For qualitative comparison against strong one-step generative baselines, we evaluate MeanFlow and StyleGAN-XL on ImageNet-256. For MeanFlow, we use the PyTorch implementation and publicly released checkpoint[1] , which achieves a slightly better FID than reported in the original paper (3.39 FID compared to originally reported 3.43 FID (Geng et al., 2026)). As shown in Fig. 18-23, our method produces samples with noticeably higher fidelity than both baselines. For a fair comparison, we use a truncation value of 0.3 for both our model and StyleGAN-XL/2, while MeanFlow is trained by a guided flow field.

---

[1]https://github.com/zhuyu-cs/MeanFlow

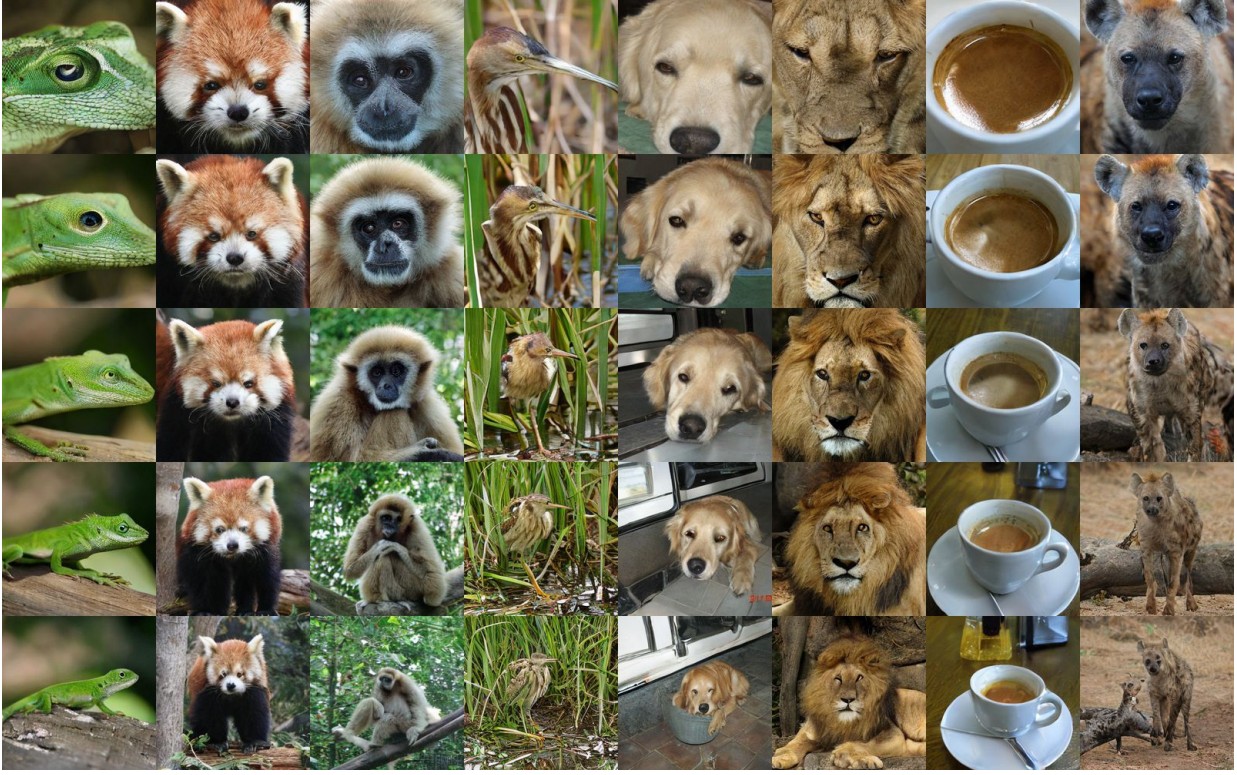

*Figure 16.* Image editing by GANSpace (Härkönen et al., 2020). We modify the top-2 principal components in the $W$ space, which produces a smooth zooming effect in the generated images.

"Sleeping"         "Roaring"         "Happy, smiling"

"in a snow field"       "filled with coffee"    "yellow lemon, replacing orange"

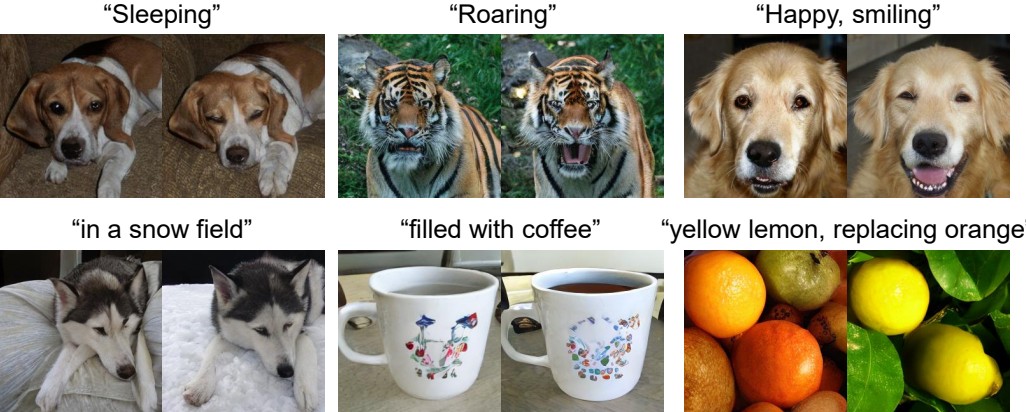

*Figure 17.* Image editing by StyleCLIP (Patashnik et al., 2021). The generated images faithfully follow the given text prompts, demonstrating that the edits successfully capture the desired semantics.

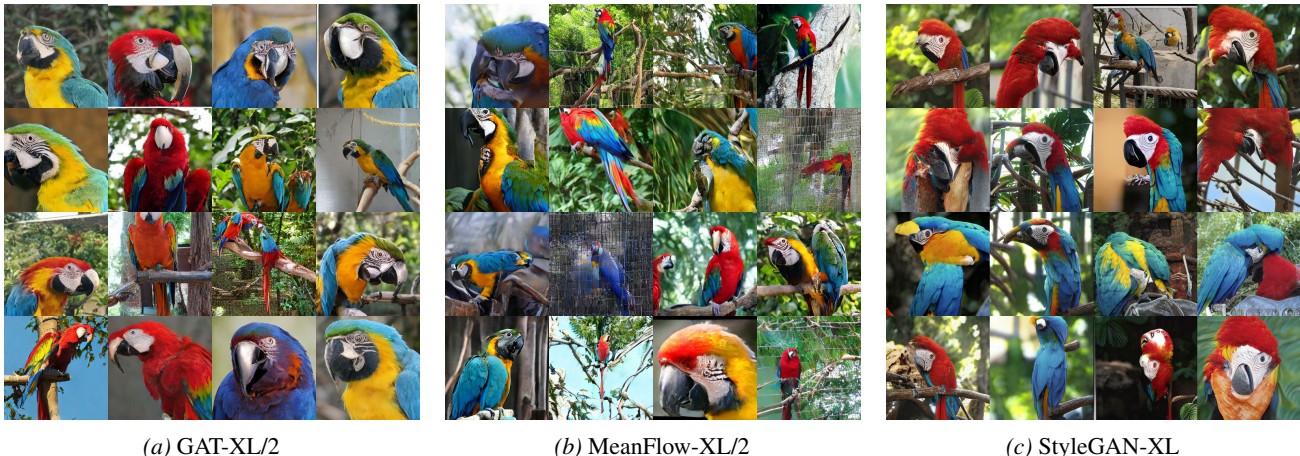

*(a)* GAT-XL/2        *(b)* MeanFlow-XL/2        *(c)* StyleGAN-XL

*Figure 18.* Qualitative comparison on ImageNet-256 by uncurated examples with 1-step generative models (MeanFlow (Geng et al., 2026) and StyleGAN-XL (Sauer et al., 2022), Class 88).

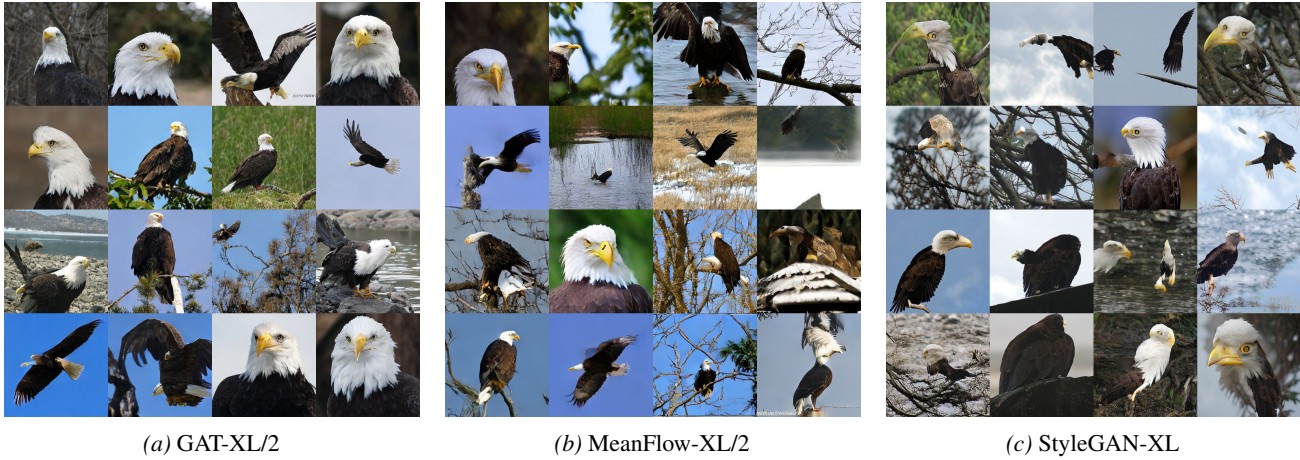

*(a)* GAT-XL/2        *(b)* MeanFlow-XL/2        *(c)* StyleGAN-XL

*Figure 19.* Qualitative comparison on ImageNet-256 by uncurated examples with 1-step generative models (MeanFlow (Geng et al., 2026) and StyleGAN-XL (Sauer et al., 2022), Class 22).

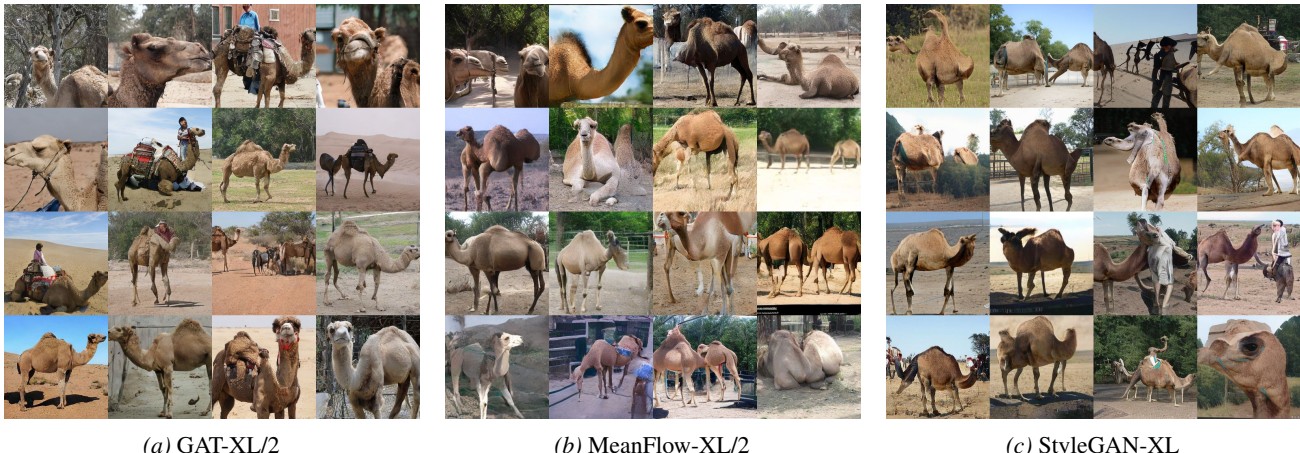

*(a)* GAT-XL/2        *(b)* MeanFlow-XL/2        *(c)* StyleGAN-XL

*Figure 20.* Qualitative comparison on ImageNet-256 by uncurated examples with 1-step generative models (MeanFlow (Geng et al., 2026) and StyleGAN-XL (Sauer et al., 2022), Class 354).

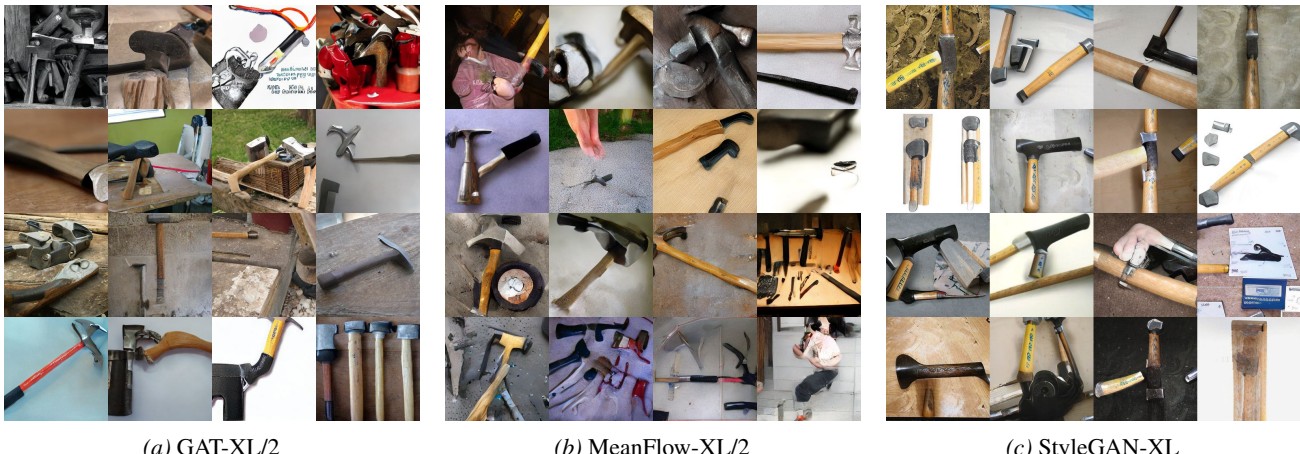

*(a)* GAT-XL/2          *(b)* MeanFlow-XL/2          *(c)* StyleGAN-XL

*Figure 21.* Qualitative comparison on ImageNet-256 by uncurated examples with 1-step generative models (MeanFlow (Geng et al., 2026) and StyleGAN-XL (Sauer et al., 2022), Class 587).

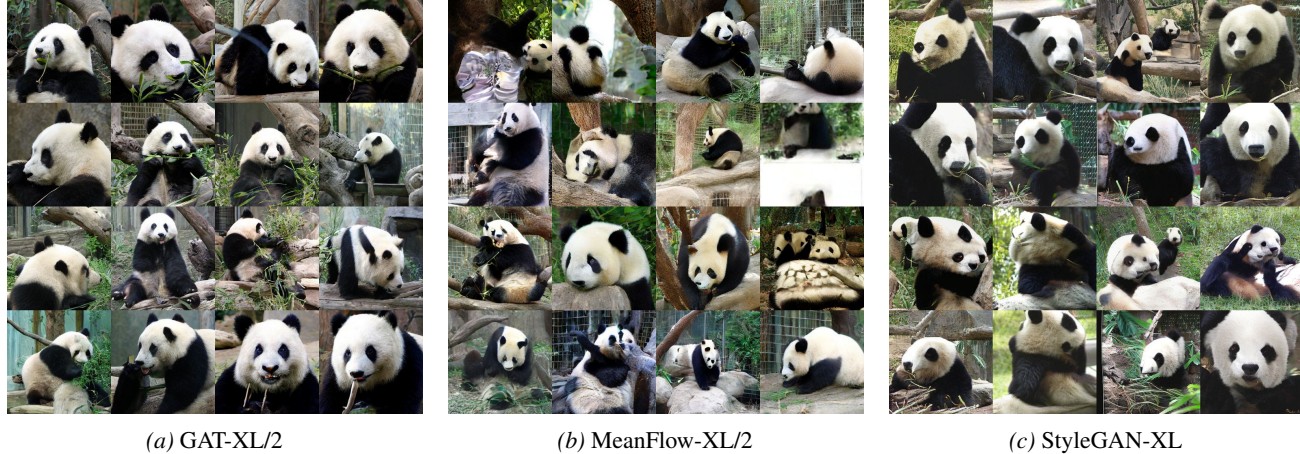

*(a)* GAT-XL/2          *(b)* MeanFlow-XL/2          *(c)* StyleGAN-XL

*Figure 22.* Qualitative comparison on ImageNet-256 by uncurated examples with 1-step generative models (MeanFlow (Geng et al., 2026) and StyleGAN-XL (Sauer et al., 2022), Class 388).

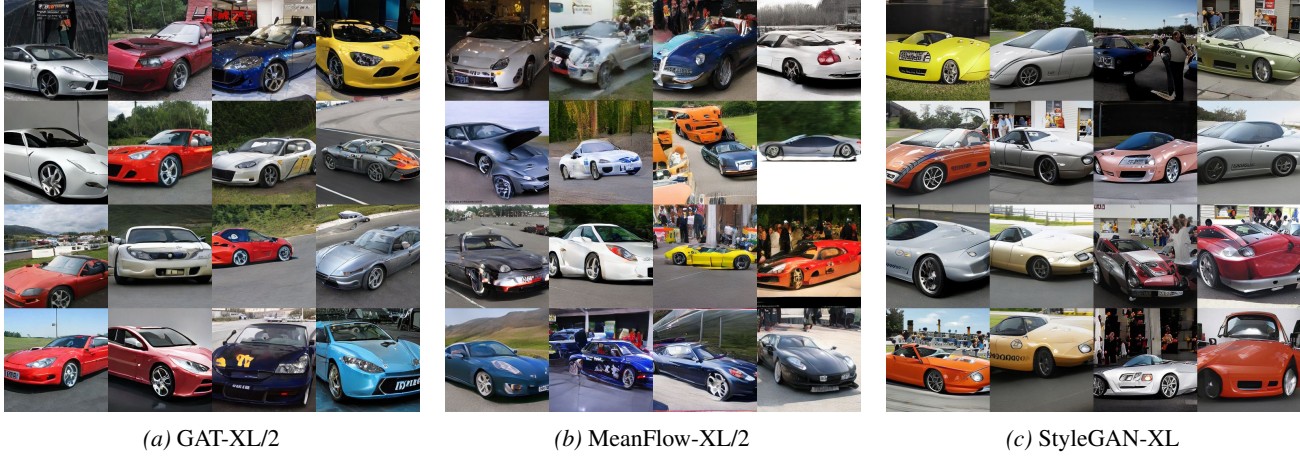

*(a)* GAT-XL/2          *(b)* MeanFlow-XL/2          *(c)* StyleGAN-XL

*Figure 23.* Qualitative comparison on ImageNet-256 by uncurated examples with 1-step generative models (MeanFlow (Geng et al., 2026) and StyleGAN-XL (Sauer et al., 2022), Class 817).

*Table 9.* Robustness of GAT to different image tokenizers on ImageNet-256. Both tokenizers use a downsampling ratio of 8.

| Tokenizer | Latent dim | Model | Epochs | FID-50K |
|---|---|---|---|---|
| SD-VAE | 4 | GAT-L/2 | 20 | 4.60 |
| FLUX-e2e (Leng et al., 2025) | 16 | GAT-L/2 | 20 | 3.73 |

## A.14. Experiments on a different tokenizer

We further assess the robustness of our framework to the choice of image tokenizer by training GAT on latents produced by an alternative encoder. Specifically, we encode ImageNet-256 using the recent FLUX-e2e tokenizer (Leng et al., 2025), which produces 16-dimensional latents with the same downsampling ratio of 8 as SD-VAE. In this latent space, we train a GAT-L/2 model for 20 epochs and obtain an FID-50K of 3.73, which surpasses the performance achieved with SD-VAE latents after 20 epochs of training (Table. 9). This result indicates that the proposed GAT framework is robust to the tokenizer choice and can potentially benefit even further from advances in image tokenizers.

## A.15. Formal scaling law

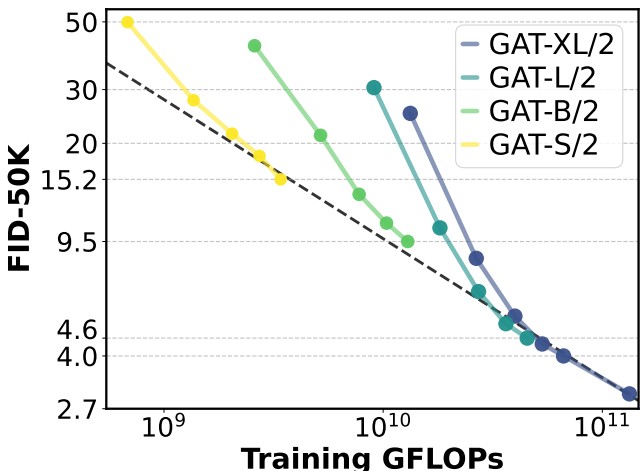

*Figure 24.* Compute–FID scaling of GAT models. We plot all training checkpoints of GAT-S/2, B/2, L/2, and XL/2 on a log–log plane, with total training compute $C$ (GFLOPs) on the x-axis and FID-50K on the y-axis. The dashed line shows a power-law fit over one summary point per model, following $\text{FID}(C) \approx 3.52 \times 10^5 \cdot C^{-0.456}$.

Fig. 24 analyzes the scaling behavior of our models with respect to training compute. We plot all training checkpoints for each model as trajectories on a log–log compute–FID plane, where the x-axis denotes the total training compute $C$ in GFLOPs (FLOPs per iteration $\times$ number of iterations with batch size 512, including G, D, VAE decoding, and the approximated GP), and the y-axis reports FID-50K. On top of these trajectories, we fit a power law using a single summary point per model, namely the final-iteration FID of GAT-S/2, B/2, L/2, and XL/2 (for GAT-XL/2 we use the 100K-iteration checkpoint reported in this paper and others for 50K-iteration). This yields the empirical relation

$$\text{FID}(C) \approx 3.52 \times 10^5 \cdot C^{-0.456}, \tag{23}$$

indicating a smooth, approximately power-law improvement of FID with training compute, consistent with scaling trends observed in diffusion and autoregressive models.

## A.16. Robustness across random seeds

Due to the high computational cost, it was challenging to run exhaustive multi-seed experiments for all configurations. To get a rough sense of seed sensitivity, we trained GAT-S/2 on ImageNet-256 for 10 epochs (25K iterations) with three different random seeds. The resulting FID-5K scores were 31.157, 28.907, and 31.069, compared to 30.085 for the originally reported run. These results suggest that the performance is not highly sensitive to the choice of random seed.

*Table 10.* Effect of VFM alignment (REPA) on FID-5K for different model sizes.

| Model | Epochs | REPA | FID-5K |
|-------|--------|------|--------|
| GAT-S/2 | 10 | | 51.43 |
| GAT-S/2 | 10 | ✓ | 30.09 |
| GAT-S/2 | 20 | | 38.99 |
| GAT-S/2 | 20 | ✓ | 22.08 |
| GAT-B/2 | 10 | | 35.67 |
| GAT-B/2 | 10 | ✓ | 23.07 |

## A.17. Detailed analysis of VFM alignment objective

**Computational overhead** In this configuration (batch size 512, 4×RTX A6000 GPUs), computing the VFM alignment (REPA) term adds about 166 ms per iteration, corresponding to under 10% of the wall-clock time per training step even for the GAT-S/2 model. This overhead remains modest because the VFM encoder (DINOv2-B/16) is frozen and used only in forward mode (no backpropagation through the teacher), and the alignment is applied only to real images, not to generated samples. If this cost is still a concern, one can precompute and cache the teacher features for all real images before training, in which case the runtime cost of the VFM alignment during GAN training is almost negligible (only a lightweight projection and similarity computation remain).

**Effectiveness of VFM alignment across various model size** We initially evaluated the effect of the VFM alignment objective (Eq. 10) through an ablation study on the GAT-S/2 model in the manuscript 5c. As summarized in Table 10, adding REPA consistently improves FID-5K: at 10 epochs, it reduces FID-5K from 51.43 to 30.09, and at 20 epochs, from 38.99 to 22.08. To verify that this effect is not tied to a particular model size, we additionally conduct an ablation on GAT-B/2. After training GAT-B/2 for 10 epochs without the VFM alignment term, the FID-5K degrades to 35.67, compared to 23.07 when the VFM alignment objective is used.

**Ablation of VFM alignment when fine-tuning the GAT** We explicitly tested this scenario by fine-tuning a GAT-B/2 model after pretraining with VFM alignment. Starting from a 50K-iteration checkpoint trained with VFM alignment, we continued training for an additional 10K iterations without the VFM alignment term. In this setting, FID-5K increased slightly from 15.6 to 17.7, indicating that removing alignment does not cause catastrophic training collapse, but does lead to a moderate degradation in performance. We interpret this as the discriminator gradually losing the semantically meaningful features acquired during VFM alignment pretraining and thus providing weaker gradients, which is consistent with observations from self-supervised GANs (Chen et al., 2019) where discriminator features erode under prolonged adversarial training alone.

We expect that, even without VFM alignment, similar effects of losing features could be mitigated via alternative regularizers such as self-supervision (Oquab et al., 2024) or distilling generator features into the discriminator (e.g., GGDR-style objectives (Lee et al., 2022a)), and we regard a systematic study of these alternatives as an interesting direction for future work.

## A.18. Additional visualizations

In the following, we provide additional visualizations of our model. The section comprises parts as belows:

- Generated samples across model scales (20 epochs).

- Latent interpolation examples from GAT-XL/2.

- PCA visualizations of intermediate features from GAT-XL/2.

- Additional generation results from GAT-XL/2.

## A.19. Generated examples from models with various scales (20 epochs)

We provide uncurated examples generated from models with various scales.

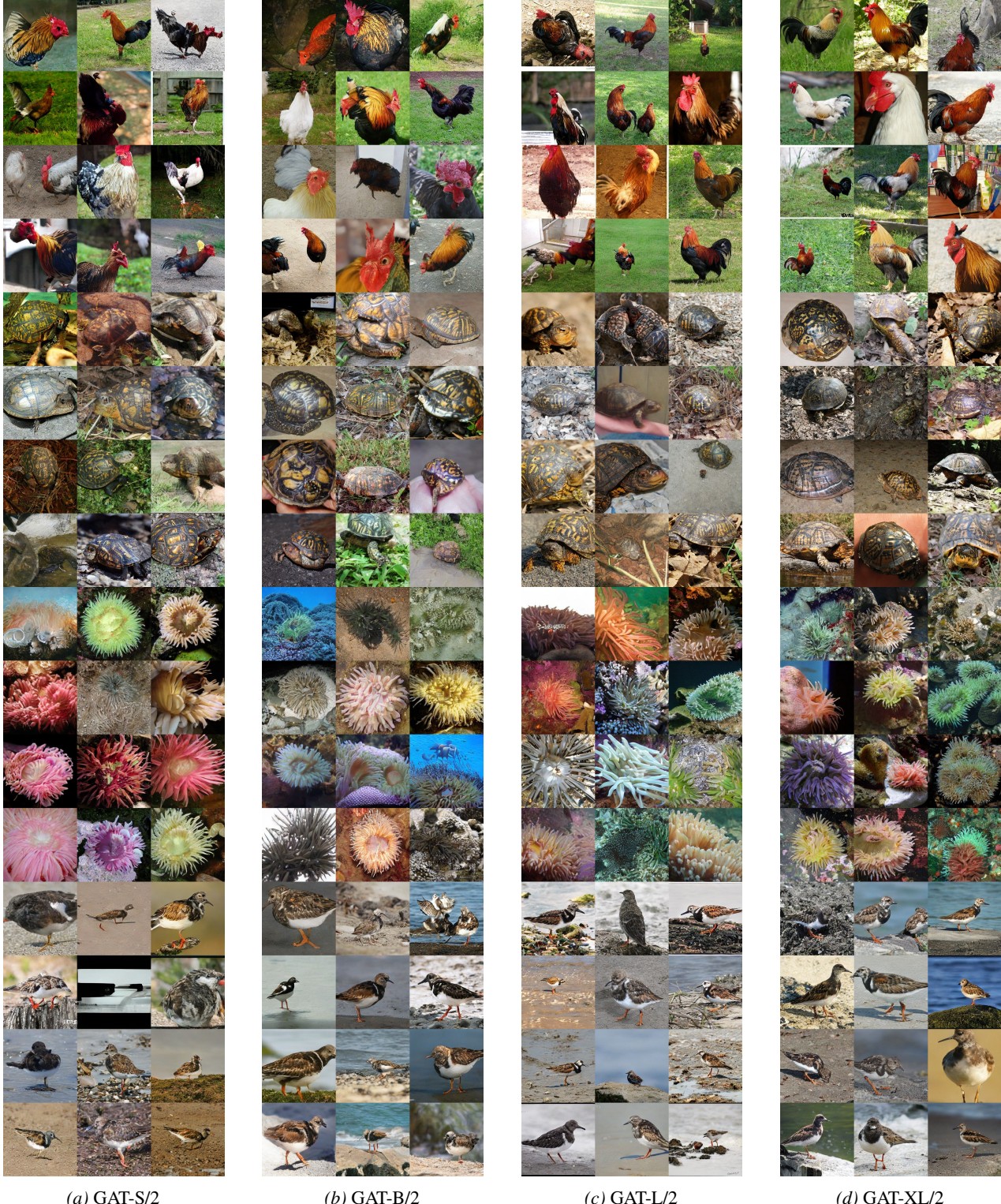

|           |           |           |            |
| :-------: | :-------: | :-------: | :--------: |
| (a) GAT-S/2 | (b) GAT-B/2 | (c) GAT-L/2 | (d) GAT-XL/2 |

*Figure 25.* Uncurated examples across model scales. From left to right, model size increases from GAT-S to GAT-XL. All models are trained for 50K iterations (i.e., 20 epochs).

### A.20. Latent interpolation examples (GAT-XL/2)

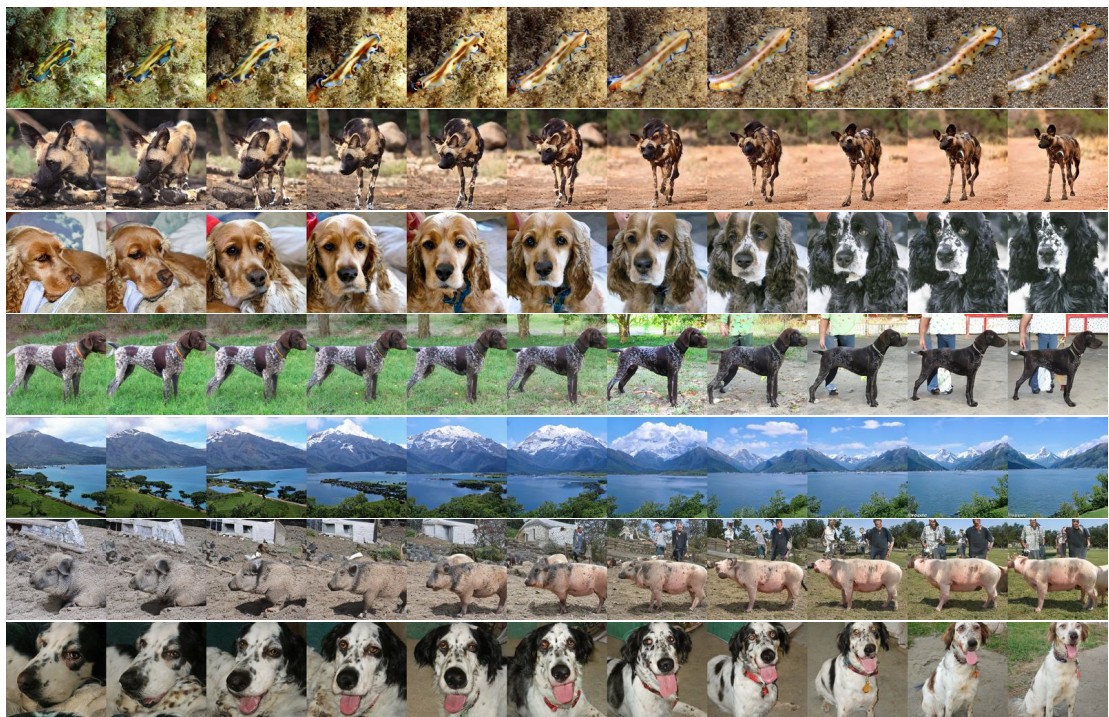

*Figure 26.* Latent interpolation examples between intra-class images.

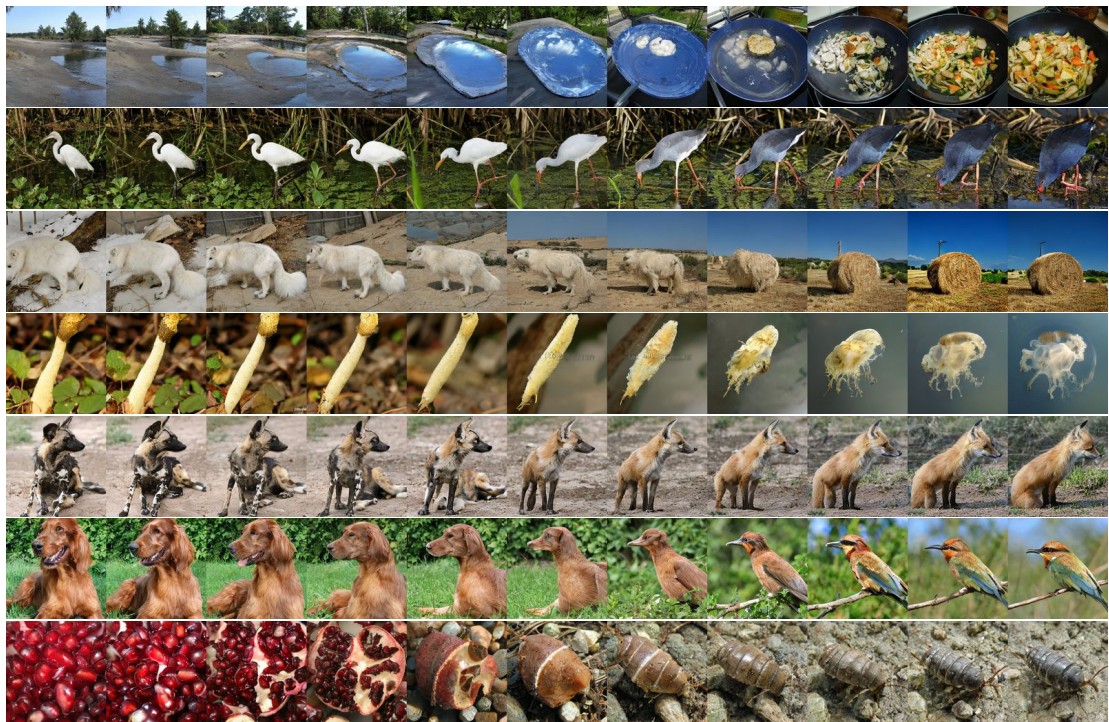

*Figure 27.* Latent interpolation examples between inter-class images.

**A.21. Visualization of intermediate features of $G$ and $D$ (GAT-XL/2)**

We visualize intermediate features of $G$ and $D$ (GAT-XL/2) by projecting onto the top-3 PCA components. Visualizations are taken from every other block, with rows ordered as: image, feature, and attention map.

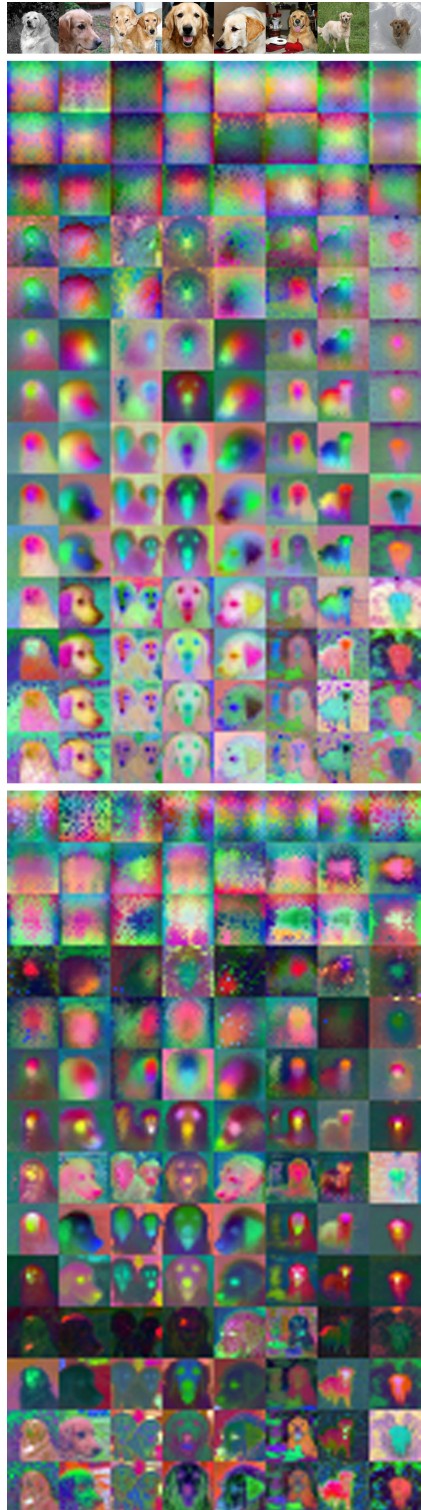

*Figure 28.* Feature visualization of $G$.

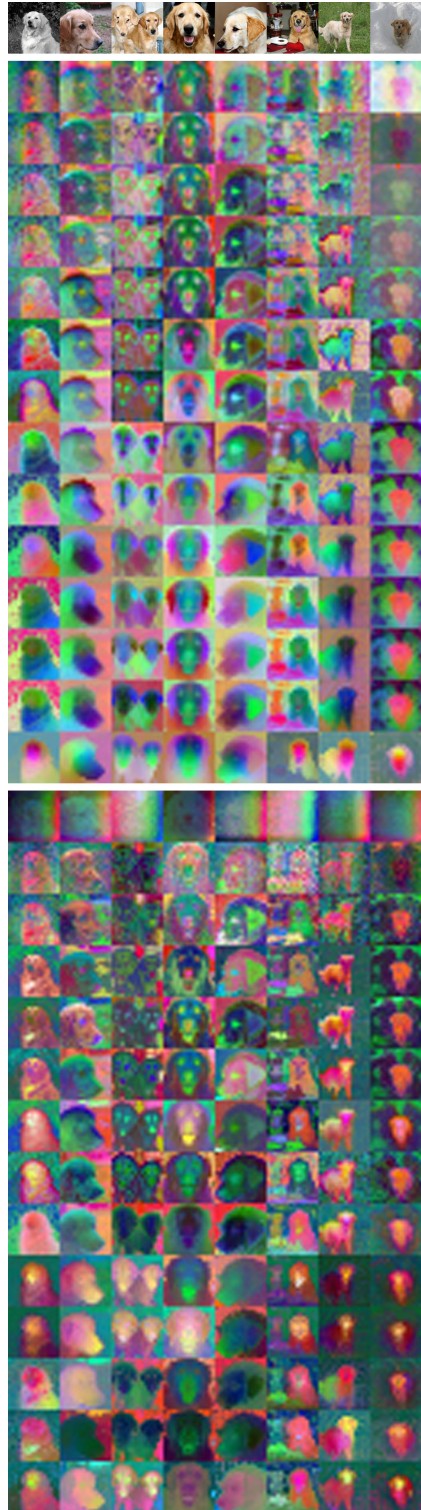

*Figure 29.* Feature visualization of $D$.

## A.22. Additional qualitative examples

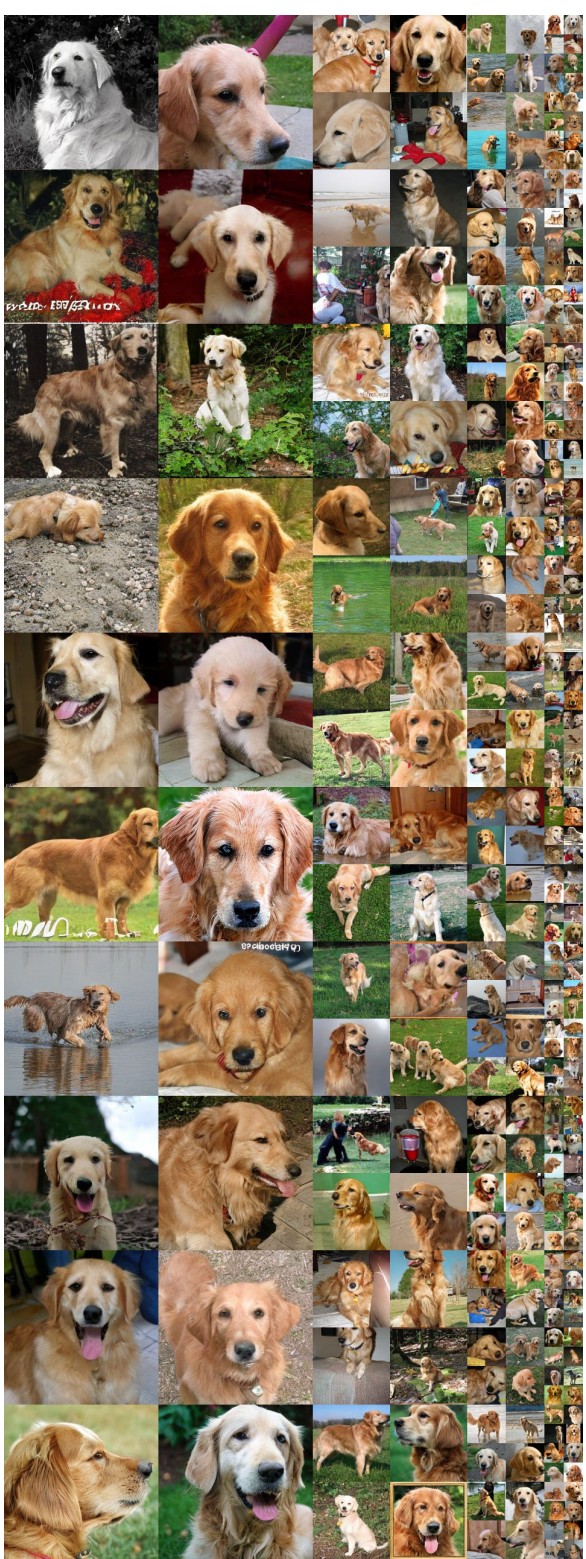

*Figure 30.* Uncurated examples from GAT-XL/2 (40 epochs). Class 207, truncation $\psi$=0.85

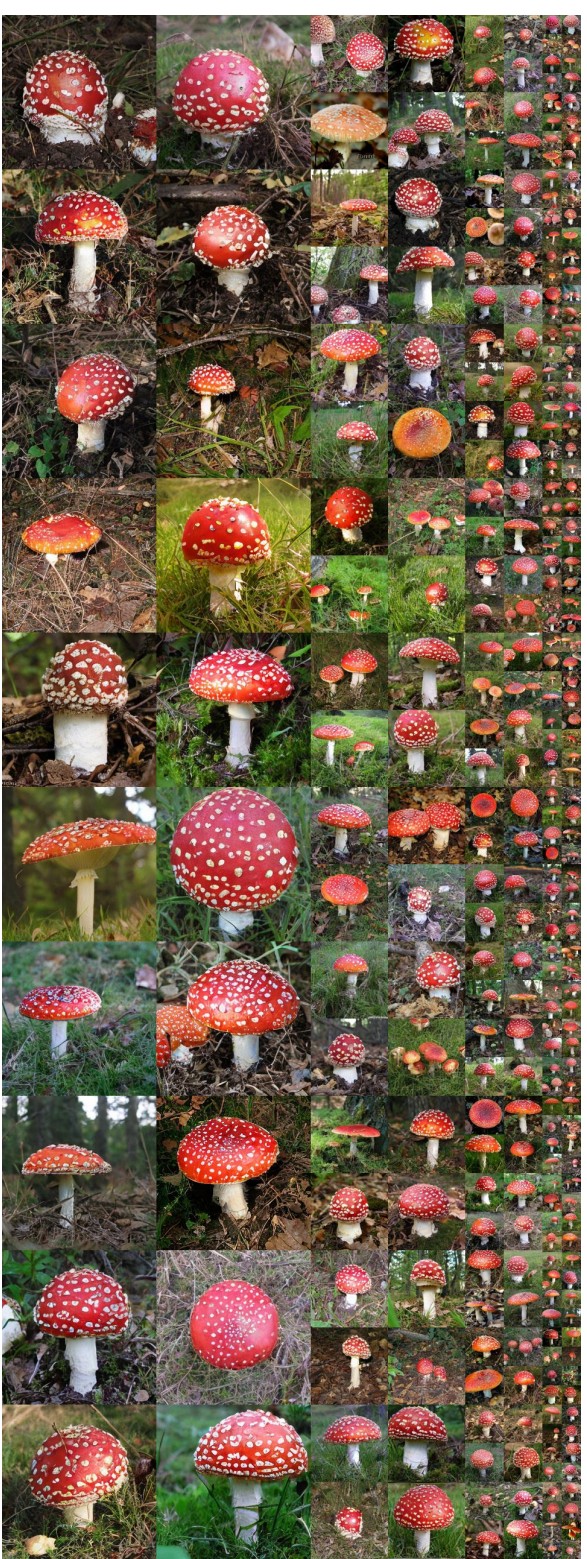

*Figure 31.* Uncurated examples from GAT-XL/2 (40 epochs). Class 992, truncation $\psi$=0.85

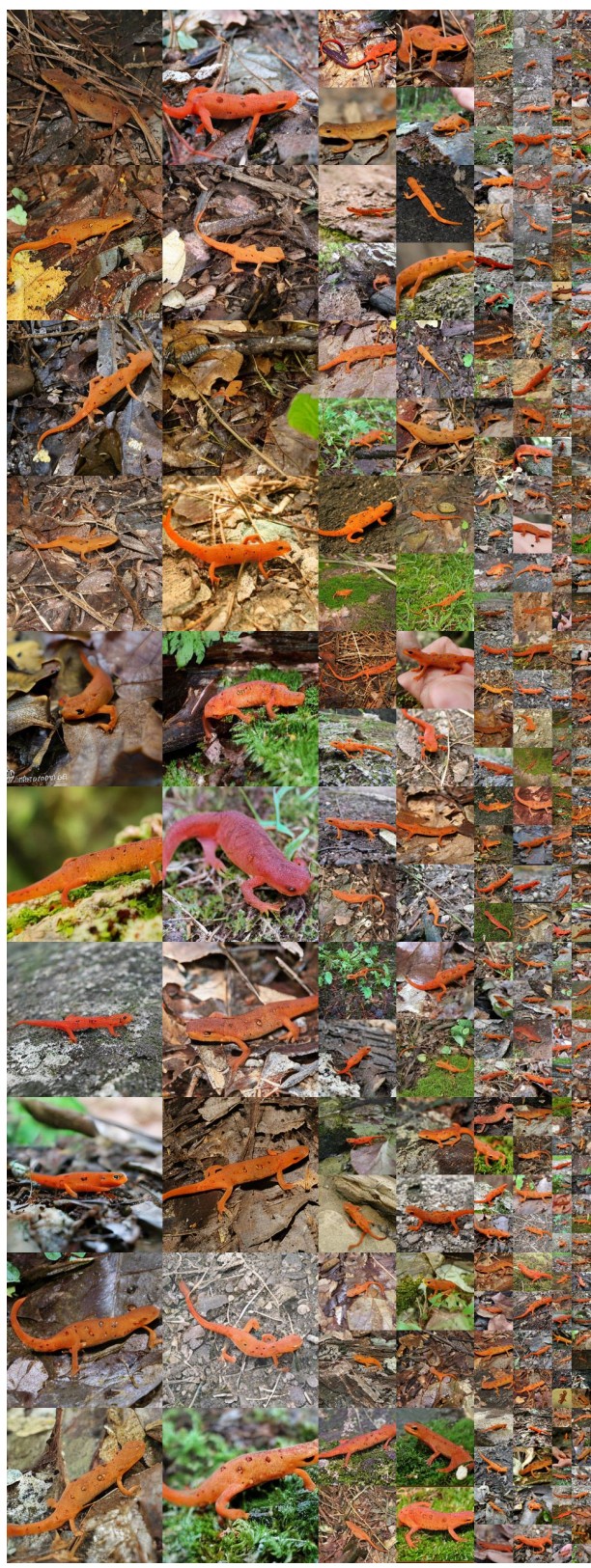

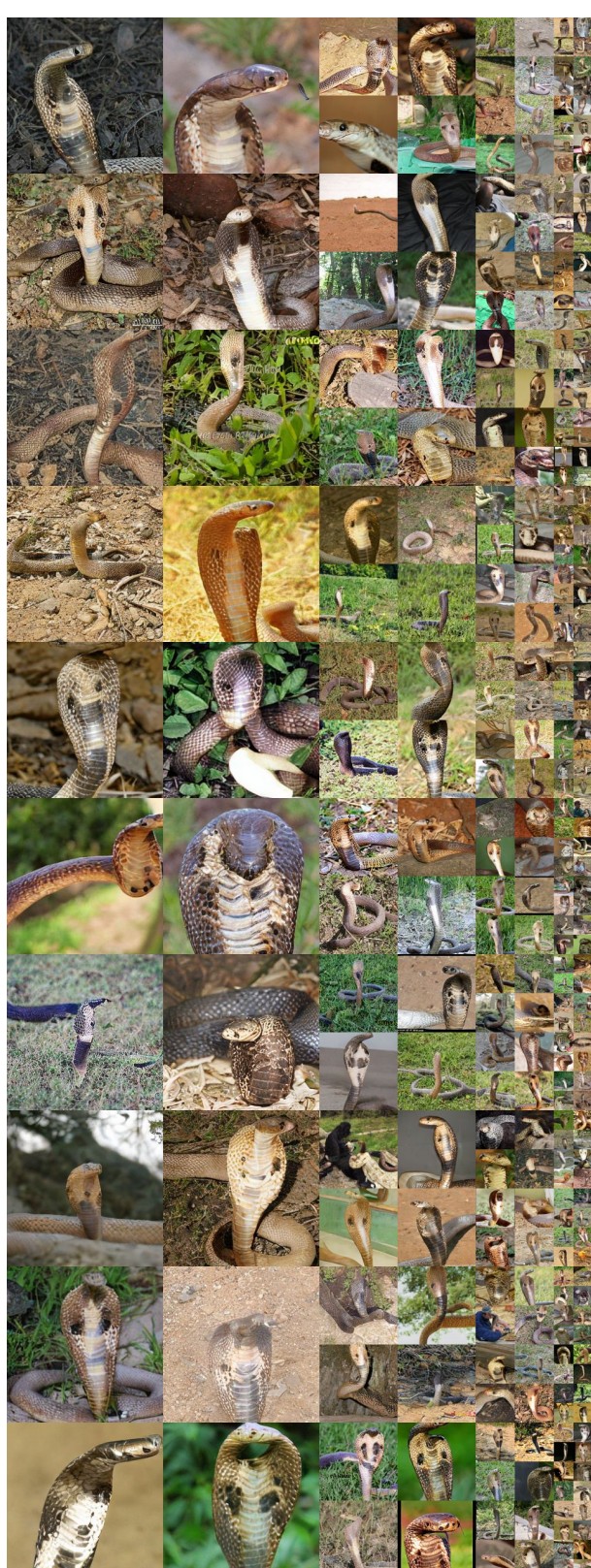

*Figure 32.* Uncurated examples from GAT-XL/2 (40 epochs). Class 27, truncation $\psi$=0.85

*Figure 33.* Uncurated examples from GAT-XL/2 (40 epochs). Class 63, truncation $\psi$=0.85

