# OpenReview forum: "Scalable GANs with Transformers"
_ICML.cc/2026/Conference — ICML 2026 regular_

### Official Review · Reviewer_hDz1 · 2026-03-02

**Soundness:** 3
**Presentation:** 3
**Significance:** 2
**Originality:** 2
**Overall Recommendation:** 4
**Confidence:** 3

**Summary:**

This paper explores a pure-transformer-based GAN architecture in image synthesis, as transformer is widely proved to exhibit good scalability in many other models and tasks yet it is underexplored in GAN. This paper therefore proposes Generative Adversarial Transformer (GAT), and studies its scaling behavior. The authors propose MNG to solve the inactive early layer issues. The overall training objective is a combination of GAN, gradient penalty and REPA. Experimental results indicate that GAT is a strong alternative to existing SOTA models, and it converges much faster.

**Compliance With Llm Reviewing Policy:**

Affirmed.

**Final Justification:**

All my major concerns and questions are properly addressed, so I raise my rate. But the authors need to make sure the name of GAT is changed and not to confuse with prior works

**Key Questions For Authors:**

1. Line 34-35 in abstract, "... naively scaling exposes failure modes", what do you mean by failure modes?
2. MNG is designed to solve the inactive early layer issues. But do you have any insights on why they will be inactive?
3. Why the proposed method can converge much faster (at least 4x faster) than other methods?
4. If you train with more epochs, how will the performance change? For example, if GAT-XL/2 is also trained with 240 epochs, will you have better or worse models?

**Limitations:**

yes

**Strengths And Weaknesses:**

## Strengths
1. The experimental results look nice, with both qualitative and quantitative visualizations. Particularly, it achieves SOTA performance among 1-NFE GANs in terms of FID.
2. Ablation and analysis are quite complete.
3. Besides proposing a new architecture and achieves SOTA performance, this work also studies the scaling effect of generator and discriminator in GAN, given they are transformer-based, which might provide valuable insights to other researchers.

## Weaknesses
1. The technical novelty is a bit limited. Besides MNG part, the rest of the design is a combination of existing techniques, such as GAN, transformer, training in the latent space (such as StableDiffusion), REPA.
2. The combination of GAN and transformer is not new, and I found that the discussion and comparison with related work is not adequate. For example, I just did a quick search and found two related works [1,2] are not included and discussed. Particularly, the title of [1] is exactly the same as your method, so I suggest considering changing the name of GAT. [2] not only did generation on 256x256, but also on 1024x1024, which seems to be a more scalable approach. What is the advantage of the propose method compared to theirs? Another related paper [3] is cited, but not compared. The current Table 1 does not contain any other transformer-based GAN. I found that your FID is still better than most of these works, so adding them into the comparison may help you.
3. The justification of MNG is not very clear. I don't quite understand the MNG part. Specifically, I don't know why adding noise, compared to no noise addition (simply add up the intermediate outputs), is necessary or more beneficial, and why adding noise to intermediate outputs will activate early layers.


[1] "Generative Adversarial Transformers", https://arxiv.org/pdf/2103.01209, ICML 2021

[2] "StyleSwin: Transformer-based GAN for High-resolution Image Generation", https://arxiv.org/pdf/2112.10762, CVPR 2022

[3] "Transgan: Two pure transformers can make one strong gan, and that can scale up", NeurIPS 2021

---

> ### Author Rebuttal · Authors · 2026-03-31
>
> Supplementary figures and tables referenced in this response are available at https://github.com/anonymous-account-31818/icml2026-31818.
> ***
> # Weakness
>
> **Q1. Limited novelty; mostly combination of existing techniques**
>
> We agree that the individual components are not entirely novel in isolation. Our contribution is in diagnosing the specific failures that prevent plain-transformer GANs from scaling reliably, and in showing that a simple combination of remedies is sufficient to recover clean scaling behavior and strong one-step generation performance. In this sense, the novelty is less in each component itself than in the scalable GAN recipe and the empirical finding that GANs can scale predictably with compute.
>
> **Q2. Discussion and comparison with transformer-based GANs [R1–3]**
>
> We highlight that prior transformer-based GANs differ from ours in a fundamental architectural sense. They all retain the resolution hierarchy of conventional ConvNets (e.g., varying spatial resolution, channel width, and attention configurations across stages) by buliding architecture that simply replaces convolutions with attention blocks. GAT, in contrast, is a stack of identical transformer blocks at fixed spatial resolution, directly inheriting the width- and depth-scaling properties of standard ViTs. This homogeneity is what enables our systematic scaling study (S/B/L/XL) under a single training recipe.
>
> Furthermore, these works evaluate only on narrow unconditional domains without evidence of scaling to benchmarks of higher diversity. To verify this gap, we trained TransGAN [R3] and StyleSwin [R2] on ImageNet-256 with increased batch size 512 for fairness. Both failed to converge (FID-50K: TransGAN 228.62, StyleSwin 178.95), while GAT-B and GAT-XL achieved 41.75 and 25.07 under the same budget (5M training images). Also, we consider renaming our method to avoid confusion with [R1].
>
> **Q3. Necessity of noise-based degradation**
>
> To clarify, early layer activation comes from the additional supervision on intermediate outputs, not from the noise itself. For this explicit intermediate supervision, we require defining what a "mid-generation sample" should look like. Without any degradation, the real counterpart at every stage is simply the repeated clean image, forcing the generator to produce collapsed and identical outputs across stages.
>
> The most straightforward degradation is spatial downsampling, but this creates a deterministic one-to-one mapping between scales. The discriminator then exploits the consistency of low- and high-resolution as a shortcut, rather than discriminating by their realness [R4]. MNG instead uses noise-based degradation, where a single clean image corresponds to many noisy variants. This one-to-many mapping prevents the discriminator from relying on cross-scale consistency.
>
> We also note that this design is particularly natural for plain transformer generator, where all blocks share the same spatial resolution.
> ***
> # Questions
> **Q4. "Failure modes" mentioned in the abstract**
>
> We refer to two specific issues: (1) early generator layers become inactive, and (2) naively applying the same learning rate across model sizes leads to instability.
>
> **Q5. Insights on why early layers become inactive**
>
> We believe this stems from the nature of GAN training. Unlike autoregressive or diffusion models, where each input is directly tied to an explicit target through next-token prediction or denoising, a GAN supervises the generator only through the discriminator's response to the final sample. This creates a natural shortcut in optimization: later layers can directly adjust the final image to satisfy the discriminator, whereas information produced by early layers becomes useful only if later layers learn to preserve and refine it. As a result, the front of the generator receives a weaker and less consistent incentive to learn meaningful structure, and can remain underutilized. This interpretation is consistent with our observations that, without MNG, early blocks have much smaller effect on the final output and are substantially less responsive to input noise.
>
> **Q6. Faster convergence of GAT compared to other methods**
>
> We attribute this to the nature of the training objective: diffusion models supervise denoising across many timesteps, so the training signal is distributed and only indirectly affects the final image. The GAN discriminator, in contrast, directly evaluates final image realism, providing a more concentrated learning signal per step. A formal analysis of this efficiency gap is an interesting future direction.
>
> **Q7. Performance when training more epochs.**
>
> We extended training of GAT-XL/2 to 80 epochs further improved FID-50K from 2.18 to 2.10, suggesting continued gains with longer schedules.
>
> ***
>
> **References**
>
> [R1] Generative Adversarial Transformers, ICML 2021.
>
> [R2] StyleSwin, CVPR 2022.
>
> [R3] TransGAN, NeurIPS 2021.
>
> [R4] Anycost GANs for interactive image synthesis and editing, CVPR 2021.

---

> > ### Author Rebuttal · Reviewer_hDz1 · 2026-04-03
> >
> > Thanks for the detailed response, which cleared all my concerns. Therefore, I raise my score to 4.

---

> > > ### Author Response · Authors · 2026-04-06
> > >
> > > We sincerely thank Reviewer hDz1 for the constructive review and for confirming that all concerns have been fully resolved. Your suggestions on transformer-based GAN comparisons and the justification of MNG led to important additions, including the TransGAN/StyleSwin convergence experiments and the clarified explanation of noise-based degradation.
> > >
> > > We will carefully revise the manuscript and these results are well reflected in the final version.

---

### Official Review · Reviewer_nA5C · 2026-03-11

**Soundness:** 3
**Presentation:** 3
**Significance:** 3
**Originality:** 3
**Overall Recommendation:** 4
**Confidence:** 3

**Summary:**

This paper proposes a new, pure transformer-based modeling approach for Generative Adversarial Networks. The authors identify two primary obstacles in scaling GANs: 1) the under-utilization of early generator layers and 2) optimization instability when increasing model dimensions. To address these issues, the authors introduce multi-level noise perturbed image guidance for intermediate supervision with a noise hierarchy. Moreover, they propose a width-aware leraning rate adjustment rule for consistent update magnitudes across different model sizes. The resulting framework, Generative Adversarial Transformers, demonstrates strong scaling behavior across different model sizes. Specifically, GAT-XL achieves a state-of-the-art FID of 2.18 on ImageNet 256 within 60 epochs, showing high data efficiency compared to existing one-step generative models.

**Compliance With Llm Reviewing Policy:**

Affirmed.

**Final Justification:**

The rebuttal adressed my concerns and I remain my score as stated in my response.

**Key Questions For Authors:**

1. How does the specific design of the noise schedule in MNG impact final quality? A more detailed ablation on different design choices would be helpful.

2. The results show that scaling the discriminator is more effective than scaling the generator. Is there a recommended ratio of parameters between G and D for future scaling?

3. Can the width-aware learning rate rule be combined with large batch training techniques to further accelerate training?

4. What is the key benefit of GAN based approach given that there are many one-step diffusion models that are also highly efficient and perform better than GAN-based models?

**Limitations:**

yes

**Strengths And Weaknesses:**

- Strengths
1. The empirical evaluation is comprehensive, spanning various model scales and including detailed ablations for MNG and the scaling rule.
2. The MNG strategy is a novel way to address inactivity issues within transformer architectures.
3. The paper is well-structured and the figures clearly illustrate the architectural details and scaling trends.
- Weaknesses
1. While training requires fewer epochs, the per iteration cost remains higher than diffusion models due to dual updates and gradient penalties, leaving the overall cost unknown.
2. The width-aware scaling rule is empirically effective but could benefit from more formal theoretical grounding in the main text.
3. The evaluation focuses heavily on ImageNet and MS-COCO, and testing on more diverse datasets would further validate the generality of the approach.

---

> ### Author Rebuttal · Authors · 2026-03-31
>
> Supplementary figures and tables referenced below are available at https://github.com/anonymous-account-31818/icml2026-31818.
> ***
> # Weakness
>
> **Q1. Per-iteration cost higher than diffusion; overall cost unknown.**
>
> We agree that per-iteration cost is higher due to dual updates and gradient penalties. However, GAT-XL/2 achieves better FID (2.18 vs. 2.27) with roughly 2× shorter total training time than DiT-XL/2, even on less powerful hardware (8×A6000 for ours vs. 8×A100 for DiT). Details are provided in Appendix A.6.
>
> **Q2. Width-aware scaling rule lacks formal theoretical grounding in main text.**
>
> We agree; a concise version of the derivation in Appendix A.10 will be moved into the main text in the revision.
>
> **Q3. Evaluation limited to ImageNet and MS-COCO.**
>
> We chose ImageNet and MS-COCO as the most widely adopted benchmarks for class-conditional and text-to-image generation. Within ImageNet, we validated generality across four model scales (S/B/L/XL), two resolutions (256, 512), two patch sizes (2, 4), and two tokenizers (SD-VAE, FLUX-e2e). We agree that additional datasets would further strengthen the claims and consider this a valuable future direction.
>
> ***
> # Questions
>
> **Q4. Impact of the noise schedule design of MNG on quality**
>
> In addition to the linear vs. exponential comparison in Fig. 5a, we conduct a further ablation varying the decay rate of the exponential schedule in Fig. R4. Specifically, we compare three schedules for K=4 intermediate outputs: (i) **linear**, where α decreases uniformly from 1.0 to 0.25; (ii) **exp-1/2** (our default), where the signal ratio halves at each level, yielding α = (0.125, 0.25, 0.5, 1.0); and (iii) **exp-1/3**, where it reduces by a factor of three, yielding α ≈ (0.037, 0.111, 0.333, 1.0). At 50K iterations on ImageNet 256, FID-50K is **15.27** (exp-1/2), **17.66** (exp-1/3), and **18.96** (linear).
>
> The exp-1/2 schedule achieves the best performance, while exp-1/3 and linear yield comparable but lower results (Tab. R2). We attribute this to how each schedule distributes α across stages: exp-1/3 drives early-stage α too close to zero, so those outputs are dominated by noise and provide little structural guidance. Linear spacing, conversely, allocates too much capacity to fine-grained stages at the expense of coarse structure. Exp-1/2 lies between these two extremes, retaining enough clean signal in early stages for global structure while maintaining clear separation from finer stages. Visualizations are provided in Fig. R3–4 at the attached link.
>
> **Q5. Recommended parameter ratio between G and D**
>
> We define an intermediate model size M (C=576, 9 heads) and compare symmetric (G-M/D-M) and asymmetric (G-S/D-B, G-B/D-S) configurations under a near-matched compute budget (Tab. R2). The symmetric G-M/D-M configuration achieves competitive FID of 11.47 to either asymmetric counterpart (FID of 11.45 and 12.62), despite requiring slightly fewer total parameters and GFLOPs, suggesting that balanced scaling may be the most compute-efficient strategy when the budget permits.
>
> **Q6. Combination of width-aware LR with large batch training**
>
> To evaluate compatibility with large-batch training, we trained GAT-S/2 with a 4× larger batch size, applying the square root scaling rule [R1]. Under this setting, the model reaches comparable performance (FID-50K of 15.86 compared to 15.27 of original model) in 1/4 of the iterations (Fig. R5), confirming that our width-aware LR transfer rule composes naturally with standard batch-size scaling. We believe this constitutes strong evidence for the practical scalability of our framework.
>
> **Q7. Key benefit of GANs compared to one-step diffusion models**
>
> Beyond competitive FID, GANs offer distinctive advantages over one-step diffusion models. First, GANs learn a semantically structured latent space supporting smooth interpolation (Fig. 9) and off-the-shelf editing (Fig. 15, 16). Second, GANs require no reconstruction objective, making them naturally suited for settings with only unpaired 2D supervision, most notably 3D-aware generation [R2], where diffusion-based alternatives rely on score distillation (SDS) and suffer from well-known quality degradation such as oversaturation and oversmoothing [R3]. We believe this work is valuable as it preserves these unique advantages while closing the quality gap with diffusion models.
>
> ***
>
> **References**
>
> [R1] Malladi, Sadhika, et al. "On the SDEs and scaling rules for adaptive gradient algorithms," NeurIPS 2022.
>
> [R2] Chan et al., "Efficient geometry-aware 3D generative adversarial networks," CVPR 2022.
>
> [R3] Wang et al., "ProlificDreamer: High-fidelity and diverse text-to-3D generation with variational score distillation," NeurIPS 2023.

---

> > ### Author Rebuttal · Reviewer_nA5C · 2026-04-01
> >
> > The reviewer thanks the authors for the thorough rebuttal with additional experiments. The responses to the reviewer's questions are satisfactory, particularly the noise schedule ablation (Q4), the G/D parameter ratio study (Q5), and the large batch compatibility experiment (Q6). The argument regarding GAN's unique advantages over one-step diffusion models (Q7) is also appreciated.
> >
> > However, after reading other reviewers' comments, the reviewer shares some of the concerns raised regarding the limited technical novelty beyond MNG (as noted by Reviewer hDz1) and the overall training complexity involving multiple loss terms and stabilization mechanisms (as noted by Reviewer tCLq). While the authors demonstrate that hyperparameters transfer across scales, the number of interacting components remains nontrivial.
> >
> > Given these considerations, the reviewer maintains the original score.

---

> > > ### Author Response · Authors · 2026-04-06
> > >
> > > We sincerely thank Reviewer nA5C for the thorough review and for acknowledging that all concerns have been fully resolved. Your suggestions directly led to valuable additions, including the noise schedule ablation, the G/D ratio study, and the large batch compatibility experiment and arguments about advantages of GANs over the other generative models, which we believe have meaningfully improved the paper.
> > >
> > > We will carefully revise the final manuscript by integrating these results.

---

### Official Review · Reviewer_tWZN · 2026-03-13

**Soundness:** 2
**Presentation:** 3
**Significance:** 2
**Originality:** 2
**Overall Recommendation:** 4
**Confidence:** 4

**Summary:**

The authors propose a method to train a GAN using a transformer architecture. To address the natural instability of GAN training, they introduce several contributions. First, they use Multi-Level Noise-Perturbed Image Guidance, which encourages the model to use every layer and results in a diffusion-like objective during training. They also scale down the learning rate as the model size increases. Finally, they distill DINO features into the discriminator.

**Compliance With Llm Reviewing Policy:**

Affirmed.

**Final Justification:**

Overall, the rebuttal + the additional rebuttal to other reviewers clarified my main concerns, and I revise my assessment to a weak accept.

**Key Questions For Authors:**

- Multi-noise looks like a diffusion objective. More intuition would be nice, along with ablations on the noise scheduler, as well as some visualization during sammpling.

- Diffusion or flow matching methods are slower but easier to monitor during training. In your case, does the validation loss correlate in some aspect with the FID? Is the training stable? Some variance on the score across different runs would help to convince that the method is more stable to train.

- Can the authors provide sampling details? Are truncation tricks used or some sort of CFG?

- Why rely only on FID? What about IS, Recall, Precision, and other metrics?

- GANs suffer from unstable training; could the authors provide the standard deviation across different runs to show that the method is stable enough?

**Limitations:**

yes

**Strengths And Weaknesses:**

Strengths:
- Multi-level Noise-perturbed Guidance seems like an interesting idea to stabilize or control the behavior of the internal representation of the GAN.
- Benefit from the speed of GANs and the nice disentanglement property of the latent space.
- Achieve competitive FID on ImageNet 256 with vanilla diffusion model such as DiT/SiT.

Major Weakness:
- The claim about inactivity of the earlier layers is already known ("Scaling up GANs for Text-to-Image Synthesis" or "Analyzing and Improving the Image Quality of StyleGAN"). Moreover, while the experiment shows that the early layers without MNG are inactive, it seems that with MNG, the later layers are inactive (Fig. 4a and 4b).

- A lot of ablations are done on 5k samples for the FID, which is difficult to compare against other works and not reliable, as FID usually requires ~10k to 50k samples to be consistent on ImageNet.

- The second main contribution on the learning rate is a bit weak. In practice, the learning rate is already one of the first hyperparameters to search when training a model from scratch (see "Deep Learning (CHAPTER 11)" by Goodfellow et al. and "GANs Trained by a Two Time-Scale Update Rule Converge to a Local Nash Equilibrium").

- The authors achieve better FID compared to other GAN methods but use five times the number of parameters. GAT-B/2 (116.75M params) in Fig. 3 seems to be worse than BigGAN or StyleGAN-XL with comparable parameters.

Minor Weakness:

- StyleGAN-XL achieves comparable results using an outdated architecture and much fewer parameters. The authors claim "Leveraging ImageNet-pretrained discriminators lowers FID more than the actual image quality," but GAT also uses DINO distillation(repa) for the discriminator.

- Improvement seems to plateau as the number of parameters increases from L to XL. Is the model truly scalable? Nowadays, diffusion models scale to billions of parameters for text-to-image tasks. Appendix A.8 is interesting and should be strengthened, as it does not yet support the claim of "scalability" in terms of data and parameters.

- Typo line 256 "Inferenece".

---

> ### Author Rebuttal · Authors · 2026-03-31
>
> Supplementary figures and tables are available at https://github.com/anonymous-account-31818/icml2026-31818.
> ***
> # Weakness
>
> **Q1. Novelty of inactivity of the earlier layers & inactivity of the latter layers**
>
> We emphasize that prior works focus on hierarchical ConvNets (GigaGAN, StyleGAN2) where later layers operate at higher resolution, naturally concentrating compute in later stages. In plain transformers, where all blocks share identical capacity, the early-layer inactivity we observe is an emergent property, to our knowledge, first highlighted in this work.
>
> Regarding the inactivity of latter layers, they are not inactive but rather serve a distinct role: synthesizing fine-grained texture. Fig. 12 confirms their feature-level contribution remains significant (e.g., low similarity between block input and output). Additionally, evaluating 3rd-stage intermediate outputs shows significantly degraded metrics (FID: 2.18→7.80, sFID: 5.27→9.31, IS: 245.4→178.4), confirming that later layers are primarily responsible for perceptual fidelity.
>
> **Q2. Unreliable FID in ablation (only 5K samples)**
>
> We have recomputed all ablation results with FID-50K (Fig. R2); the relative trends are consistent with our original conclusions.
>
> **Q3. Weak contribution of the width-adaptive LR**
>
> We agree that learning-rate tuning is standard practice. Our contribution addresses a different problem: hyperparameter transfer across scales. In practice, tuning is feasible at small scale but becomes prohibitively expensive at large scale. Establishing reliable transfer rules from small to large models is therefore a practically significant problem. We propose a width-dependent rule that transfers a single tuned learning rate from GAT-S (4e-4) to B/L/XL, with all scales converging stably without additional search. Fig. 5b confirms this is not redundant: naively reusing one scale's rate at another causes slow convergence or divergence.
>
> **Q4. Comparable FID, but requires a large number of parameters**
>
> We acknowledge that GAT-XL uses more parameters than StyleGAN-XL. However, the scaling law literature [R1, R2] consistently adopts compute in FLOPs, rather than parameter count, as the primary axis for evaluating model scaling, as it more directly reflects computational cost. By this measure, GAT-XL requires only 119 GFLOPs for the generator versus 1,574 for StyleGAN-XL, roughly 13x lighter. Even including VAE decoder (310 GFLOPs), GAT-XL remains approximately 4x lighter. Regarding GAT-B, at 40 epochs with truncation, GAT-B (23 GFLOPs) achieves FID 5.07, outperforming BigGAN (59 GFLOPs, FID 6.95) at less than half the compute.
> ***
> # Minor Weakness
> **Q5. Concern about pretrained features also applying to GAT (DINOv2)**
>
> The concern is specifically that StyleGAN-XL's ImageNet-classification-pretrained discriminators (EfficientNet, DeiT-S) share feature space with Inception network used for FID, creating systematic evaluation bias [R3]. Our REPA distills DINOv2 features learned through self-supervision without classification labels, and thus does not share this bias. To verify, we compute CLIP-FID following [R3], which uses CLIP-ViT as encoder, removing classification-specific bias. GAT-XL/2 achieves CLIP-FID 1.86 versus StyleGAN-XL's 2.62. Even our 40-epoch checkpoint (CLIP-FID 2.21, FID 2.96) already surpasses StyleGAN-XL, indicating that its low FID is largely attributable to evaluation bias (Tab. R1).
>
> **Q6. Plateau of performance at large scale (L→XL)**
>
> Fig. 3d shows that FID follows a power law against training compute on a log scale, consistent with scaling trends in LLMs [R1] and diffusion models [R4]. Under a power law, the L-to-XL gap naturally appears small on a linear axis (Fig. 3a), but the log-scale plot confirms that improvement continues at a predictable rate. DiT similarly established its scaling on ImageNet (S to XL) before extending to billion-scale T2I models. We believe our compute scaling law provides a comparable foundation.
> ***
> # Questions
>
> **Q7. Intuition of MNG and visualization:** Please refer to Fig. R3, 4 for visualization, and R#hDz1-Q3, nA5C-Q4 for intuition.
>
> **Q8. Correlation between loss and FID; training stability:** Adversarial loss does not directly correlate with FID. However, GAT's single-step generation allows efficient FID monitoring without multi-step sampling. Regarding stability, three independent runs of GAT-S/2 yield FID-5K of 30.38±1.27 (original: 30.09), demonstrating low variance across seeds.
>
> **Q9. Usage of truncation trick or CFG:** We do not use truncation or CFG except for visualization (ψ=0.85, unless specified).
>
> **Q10. Other metrics:** We provide CLIP-FID, FID, IS, sFID, Precision, and Recall in Tab. R1.
> ***
> **References**
>
> [R1] Scaling laws for neural language models, arXiv 2020.
>
> [R2] Training compute-optimal large language models, NeurIPS 2022.
>
> [R3] The role of ImageNet classes in Fréchet Inception Distance, ICLR 2023.
>
> [R4] Scalable diffusion models with transformers, CVPR 2023.

---

> > ### Author Rebuttal · Reviewer_tWZN · 2026-04-04
> >
> > Thank you for the detailed and thorough rebuttal. I appreciate the additional experiments and clarifications, particularly the recomputation of FID with 50K samples + evaluation metrics, and the discussion on training stability and variance. These significantly improve the soundness of the experimental evaluation. The clarification regarding compute (FLOPs) versus parameter count is also helpful.
> >
> > Some concerns remain partially addressed. In particular, the novelty of the early-layer inactivity analysis and the role of multi-level noise guidance could benefit from deeper empirical investigation, and the scalability claims are still somewhat limited in practice.
> >
> > Overall, however, the rebuttal + the additional rebutall to other reviewers clarified my main concerns, and I revise my assessment to a weak accept.

---

> > > ### Author Response · Authors · 2026-04-06
> > >
> > > We sincerely thank Reviewer tWZN for the constructive feedback and for raising the score to weak accept. Your questions on FID reliability, compute-versus-parameter framing, and scalability analysis directly motivated key improvements: the FID-50K recomputation, the CLIP-FID evaluation removing classification-specific bias, the training stability, and FLOPs versus parameter counts. We believe these additions have meaningfully strengthened the paper.
> > >
> > > We will carefully revise the final manuscript by integrating these results.
> > >
> > > Also, we kindly ask whether the official review score could also be updated to reflect the revised assessment, which can be done via the Edit button on the original review.

---

### Official Review · Reviewer_tCLq · 2026-03-16

**Soundness:** 3
**Presentation:** 4
**Significance:** 3
**Originality:** 2
**Overall Recommendation:** 4
**Confidence:** 3

**Summary:**

The paper proposes Generative Adversarial Transformers (GAT), where both the generator and discriminator are built from Vision Transformers and trained in a compact VAE latent space to enable efficient scaling. To address issues observed when scaling GANs, specifically inactive early generator layers and instability across model sizes. The authors introduce multi-level noise-perturbed guidance for intermediate supervision and a width-aware learning-rate scaling rule. Experiments on class-conditional ImageNet generation demonstrate that the approach scales effectively and achieves strong single-step generation performance compared to prior methods.

**Compliance With Llm Reviewing Policy:**

Affirmed.

**Key Questions For Authors:**

Please see Q2

**Limitations:**

Please see Q2

**Strengths And Weaknesses:**

Strengths:
* **Clear presentation**. The paper is generally well written, logically structured, and easy to follow. The motivation, architectural choices, and training strategies are clearly explained, which makes the contributions accessible.

* **Simple and principled architecture**. The proposed model largely follows a standard Vision Transformer (ViT) design with minimal modifications to adapt it for GAN training.

* **Convincing scaling results**. The experimental results show consistent improvements as model size increases, demonstrating that the proposed approach exhibits favorable scaling behavior on ImageNet.

* **Innovative solutions to GAN training challenges**. The paper proposes two thoughtful techniques to address well-known instability issues in GAN training. The multi-level noise-perturbed supervision encourages all generator layers to contribute to synthesis, mitigating early-layer inactivity. In addition, the proposed learning-rate scaling rule is intuitive and appears effective in stabilizing training across different model sizes.

Weaknesses
* **Training complexity**. The method relies on multiple loss terms and stabilization mechanisms, including adversarial loss, gradient penalties, intermediate supervision through multi-level noise perturbation, learning-rate scaling rules, and representation alignment with pretrained models. While these components appear effective, the overall training setup becomes fairly complex and involves several engineering choices that may make the method harder to reproduce and tune.

* **Limited exploration of pretrained-feature discriminators**. The paper introduces a REPA-style alignment objective that encourages the discriminator features to match those of a pretrained vision foundation model. However, this idea is closely related to the Projected GAN line of work, which directly uses pretrained networks (e.g., DINO-based features) as discriminators or feature extractors. Although the authors acknowledge this connection in the appendix, the paper does not provide experiments comparing against a Projected GAN–style discriminator. Such a comparison would help clarify whether the proposed alignment objective offers advantages over simply using pretrained features directly.

---

> ### Author Rebuttal · Authors · 2026-03-31
>
> Supplementary figures and tables referenced below are available at https://github.com/anonymous-account-31818/icml2026-31818.
> ***
> # Weakness
>
> **Q1. Complex training setup with multiple losses and stabilization mechanisms; hard to reproduce and tune.**
>
> We acknowledge that the training setup involves multiple components. However, all hyperparameters except the learning rate are shared across model sizes (Appendix A.9), and the learning rate itself is determined by a single transfer rule (Eq. 5), so the actual tuning burden is minimal. Furthermore, the same set of hyperparameters successfully transfers without additional tuning across different settings: from class-conditional ImageNet to text-to-image MS-COCO, from 256 to 512 resolution, and even to a different tokenizer (FLUX-e2e, Tab. 8). We believe this robustness across tasks, resolutions, and tokenizers demonstrates that the training recipe is practical despite involving multiple components. We will release our code upon acceptance to support reproducibility.
>
> **Q2. No comparison with Projected GAN style discriminators that directly use pretrained features.**
>
> We thank the reviewer for this suggestion. A key structural difference is that Projected GAN operates in pixel space, where pretrained networks (e.g., DINO, EfficientNet) can directly process generated images. In our framework, both G and D operate in the VAE latent space (32×32, 4-channel for 256 resolution image), which is incompatible with pretrained vision models that expect pixel-space input.
>
> Our REPA-style alignment sidesteps this constraint: it aligns the discriminator's learned representations with VFM features using only real images (which are available in pixel space), guiding the discriminator to learn semantically rich features without requiring pixel-space decoding of generated samples. As shown in Appendix A.17, this incurs under 10% wall-clock overhead and can be further reduced by precomputing teacher features.
>
> To directly test a Projected GAN-style alternative, we conducted an additional experiment. Since our MNG strategy requires the main discriminator to operate on multi-level latent outputs, we cannot replace it with a pretrained pixel-space network; we therefore adopt an auxiliary discriminator setup following Vision-aided GAN [R1]. Specifically, we decode the generator's final-stage latent output back to pixel space through the frozen VAE decoder, pass the resulting image through a frozen pretrained feature extractor, and attach a shallow auxiliary discriminator head on top of the extracted features.
> We use the feature extractor same as REPA counterpart, DINOv2-B, for this experiment.
>
> In this setting, the pretrained-feature discriminator drives the discriminator loss to near zero compared to the REPA counterpart (Fig. R1 in the attached link), meaning it can almost perfectly separate real from generated images. When the discriminator becomes this confident, gradient feedback to the generator vanishes, and the generator receives little useful learning signal to improve. As a result, training degenerates into severe mode collapse, with Recall approaching zero. We do not view this as evidence that projected or vision-aided discrimination is fundamentally unsuitable for latent GANs; rather, it suggests that, in our setting, such a design is not plug-and-play and likely requires additional balancing or careful tuning to remain stable. This is consistent with prior observations in the Projected GAN literature, where techniques such as random feature projection and careful discriminator capacity control are required to maintain training stability.
>
> In addition, routing generated latents through the VAE decoder at every training step incurred approximately 27× higher VRAM consumption compared to our standard pipeline (measured on GAT-S/2, batch size 512). These results suggest that REPA-style alignment offers a more practical and stable mechanism for leveraging pretrained visual knowledge in latent-space GANs.
>
> ***
>
> **References**
>
> [R1] Kumari et al., "Ensembling off-the-shelf models for GAN training," CVPR 2022.

---

> > ### Author Rebuttal · Reviewer_tCLq · 2026-04-07
> >
> > I appreciate the feedback provided by the authors. Given the complexity of training and high engineering requirements, I tend to maintain my original weak acceptance rating.

---

> > > ### Author Response · Authors · 2026-04-07
> > >
> > > We sincerely thank Reviewer tCLq for the thorough review and for confirming that all concerns have been fully resolved.
> > >
> > > Regarding the training complexity, we gently note that **only the learning rate changes across scales** while all other hyperparameters remain identical, transferring successfully across different datasets, resolutions, and tokenizers.
> > >
> > > We will carefully revise the manuscript and ensure all additional experiments and clarifications are well reflected in the final version.

---

### Decision · Program_Chairs · 2026-04-30

**Decision:**

Accept (regular)

**Comment:**

After the rebuttal, the submission received 4 weak accept. The rebuttal has addressed most of the concerns from the reviewers and there is a consensus of accept. AC has checked the submission, the reviews, the rebuttal, and the discussion, and AC agreed that the paper provides a solid empirical contribution to the scaling of GANs, a sub-field that has recently been overshadowed by Diffusion model. The authors have demonstrated clear state-of-the-art results for one-step generation and provided a robust recipe for others to follow. Thus accept is recommended.